# EFFICIENT NEWTON-TYPE FEDERATED LEARNING WITH NON-IID DATA

## ABSTRACT

The mainstream federated learning algorithms only communicate the first-order information across the local devices, i.e., FedAvg and FedProx. However, only using first-order information, these methods are often inefficient and the impact of heterogeneous data is yet not precisely understood. This paper proposes an efficient federated Newton method (FedNewton), by sharing both first-order and second-order knowledge over heterogeneous data. In general kernel ridge regression setting, we derive the generalization bounds for FedNewton and obtain the minimax-optimal learning rates. For the first time, our results analytically quantify the impact of the number of local examples, the data heterogeneity and the model heterogeneity. Moreover, as long as the local sample size is not too small and data heterogeneity is moderate, the federated error in FedNewton decreases exponentially in terms of iterations. Extensive experimental results further validate our theoretical findings and illustrate the advantages of FedNewton over the first-order methods.

## 1 INTRODUCTION

Owing to the great potential in privacy preservation and in lowering the computational costs, federated learning (FL) McMahan et al. (2017); Li et al. (2020a); Zhang et al. (2021) becomes a promising framework in processing large-scale tasks. However, federated learning is facing massive challenges from the heterogeneous data Zhao et al. (2018); Zhou et al. (2023); Ye et al. (2023), including both the data heterogeneity and the model heterogeneity. The data heterogeneity comes from that inputs across devices are usually sampled from heterogeneous distributions, while the model heterogeneity measures the response shift due to inconsistency between local models and the global model.

First-order approaches, including FedAvg McMahan et al. (2017) and FedProx Li et al. (2020a), share the first-order information rather than the data across devices and tolerate the heterogeneity in federated learning, while Newton-type FL methods Ghosh et al. (2020); Gupta et al. (2021); Safaryan et al. (2022); Islamov et al. (2023); Liu et al. (2023); Dal Fabbro et al. (2024); Li et al. (2023) utilized second-order information for updating federated model. To the best of our knowledge, most of existing learning guarantees for FL methods are derived in the context of optimization and focused on in-sample predictive errors only, i.e., the convergence analysis (optimization) of first-order FL Li et al. (2020b); Karimireddy et al. (2020); Pathak & Wainwright (2020); Glasgow et al. (2022) and Newton-type FL Ghosh et al. (2020); Safaryan et al. (2022); Qian et al. (2022). However, beyond the optimization, the generalization guarantees (out-sample predictive performance) are of great practical and theoretical interests for FL. Despite recent efforts and progress on the generalization for first-order algorithms Mohri et al. (2019); Yagli et al. (2020); Su et al. (2021); Yuan et al. (2022), the generalization guarantees for Newton-type FL algorithms remain elusive, especially on heterogeneous data and localized models. Therefore, a challenging problem in FL is *how to quantify the impact of heterogeneity from the generalization perspective?*

In this paper, motivated by sharing second-order information, we propose a second-order federated optimization method, named `FedNewton`. It approximates the global predictor on the entire data by utilizing the global gradient and local Hessians, improving the predictive accuracy in an efficient communications framework. We then study the statistical properties of `FedNewton`, and derive the generalization bounds with the minimax optimal rates. We conclude with experiments on simulated data and publicly available tasks that complement our theoretical results, exhibiting the computa-

tional and statistical benefits of our approach. Due to the length limit, we leave the experiment part in the appendix. We summarize our contributions as below:

**1) On the algorithmic front.** We propose a fast second-order federated learning algorithm, which improves the approximation of the centralized model while only requiring similar computational and communication costs as the first-order methods. The convergence of `FedNewton` is exponentially fast and a few communications, for example, $t \leq 2$, can approximate the global model well.

**2) On the statistical front.** To our best knowledge, in presence of both data heterogeneity and model heterogeneity, we present the optimal generalization guarantees for the first time. Our results further analytically quantify the impacts of the local sample size, the data heterogeneity, and the model heterogeneity. Especially, the federated error decreases exponentially fast in benign cases, i.e., a sufficient number of local examples and moderate data heterogeneity.

## 2 PROBLEM SETUP

In a standard framework of federated learning, there is a global parameter server and $m$ local computational clients. On the $j$-th local machine $\forall j \in [m]$, the local data $\mathfrak{D}_j = \{(\boldsymbol{x}_{ij}, y_{ij})\}_{i=1}^{|\mathfrak{D}_j|}$ is drawn from a local distribution $\rho_j$ on the joint space $\mathcal{X} \times \mathcal{Y}$. The total sample $\mathcal{D} = \bigcup_{j=1}^m \mathfrak{D}_j$ is the disjoint union of local data and corresponds to a global distribution $\rho$. For any local devices $j, k \in [m]$ and $j \neq k$, data distributions are identical $\rho_j = \rho_k = \rho$ in the homogeneous setting (iid data), while data distributions are distinct $\rho_j \neq \rho_k$ in the heterogeneous case (non-iid data).

We base our analysis on the standard non-parametric regression setup and assume that the target solution $f^*$ belongs to a reproducing kernel Hilbert space (RKHS) induced by a Mercer kernel $K : \mathcal{X} \times \mathcal{X} \to \mathbb{R}$. Mercer's theorem guarantees the kernel function admits an implicit feature mapping $K(\boldsymbol{x}, \boldsymbol{x}') = \langle \phi(\boldsymbol{x}), \phi(\boldsymbol{x}') \rangle_K$ and the norm by $\| \cdot \|_K$. The predictor can be stated as $f_{\mathcal{D}, \lambda}(\boldsymbol{x}) = \langle \boldsymbol{w}_{\mathcal{D}, \lambda}, \phi(\boldsymbol{x}) \rangle$ where $\boldsymbol{w}_{\mathcal{D}, \lambda}$ minimizes the objective on the entire data $\mathcal{D}$

$$\arg\min_{\boldsymbol{w} \in \mathcal{H}_K} \left\{ \frac{1}{2|\mathcal{D}|} \sum_{i=1}^{|\mathcal{D}|} (f(\boldsymbol{x}_i) - y_i)^2 + \frac{\lambda}{2} \|\boldsymbol{w}\|_K^2 \right\}, \tag{1}$$

where $(\boldsymbol{x}_i, y_i) \in \mathcal{D}$, and $\lambda > 0$ is the regularity parameter. The above regression problem, known as Kernel Ridge Regression (KRR), admits a closed-form solution

$$\boldsymbol{w}_{\mathcal{D}, \lambda} = (\boldsymbol{\Phi}_{\mathcal{D}}^\top \boldsymbol{\Phi}_{\mathcal{D}} + \lambda I)^{-1} \boldsymbol{\Phi}_{\mathcal{D}}^\top \boldsymbol{y}_{\mathcal{D}}, \tag{2}$$

where $\boldsymbol{\Phi}_{\mathcal{D}} = \frac{1}{\sqrt{|\mathcal{D}|}} \left[ \phi(\boldsymbol{x}_1), \cdots, \phi(\boldsymbol{x}_{|\mathcal{D}|}) \right]^T \in \mathbb{R}^{|\mathcal{D}|} \times \mathcal{H}_K$ are feature mappings on the training set $\mathcal{D}$ and $\boldsymbol{y}_{\mathcal{D}} = \frac{1}{\sqrt{|\mathcal{D}|}} (y_1, \cdots, y_{|\mathcal{D}|})^\top$ are the corresponding labels.

By averaging the local models, the simplest federated method only communicates once, known as Distributed Kernel Ridge Regression (DKRR) with the closed-form solution

$$\bar{\boldsymbol{w}}_{\mathcal{D}, \lambda} = \sum_{j=1}^m p_j (\boldsymbol{\Phi}_{\mathfrak{D}_j}^\top \boldsymbol{\Phi}_{\mathfrak{D}_j} + \lambda I)^{-1} \boldsymbol{\Phi}_{\mathfrak{D}_j}^\top \boldsymbol{y}_{\mathfrak{D}_j},$$

where $p_j$ is the weight of the $j$-th local model, which is usually set $p_j = |\mathfrak{D}_j|/|\mathcal{D}|$. Note that, $\boldsymbol{\Phi}_{\mathfrak{D}_j} = \frac{1}{\sqrt{|\mathfrak{D}_j|}} \left[ \phi(\boldsymbol{x}_1), \cdots, \phi(\boldsymbol{x}_{|\mathfrak{D}_j|}) \right]^T \in \mathbb{R}^{|\mathfrak{D}_j|} \times \mathcal{H}_K$ are local feature mappings and $\boldsymbol{y}_{\mathfrak{D}_j} = \frac{1}{\sqrt{|\mathfrak{D}_j|}} (y_1, \cdots, y_{|\mathfrak{D}_j|})^\top$ are labels on the $j$-th local train set $\mathfrak{D}_j = \{(\boldsymbol{x}_{ij}, y_{ij})\}_{i=1}^{|\mathfrak{D}_j|}, \quad \forall j \in [m]$.

The solution of KRR equation 2 can be rewritten in the Newton's method form

$$\boldsymbol{w}_{\mathcal{D}, \lambda} = \boldsymbol{w} - \boldsymbol{H}_{\mathcal{D}, \lambda}^{-1} \boldsymbol{g}_{\mathcal{D}, \lambda}. \tag{3}$$

where the gradient and Hessian matrix are defined as

$$\boldsymbol{g}_{\mathcal{D}, \lambda} := (\boldsymbol{\Phi}_{\mathcal{D}}^\top \boldsymbol{\Phi}_{\mathcal{D}} + \lambda I) \boldsymbol{w} - \boldsymbol{\Phi}_{\mathcal{D}}^\top \boldsymbol{y}_{\mathcal{D}},$$

$$\boldsymbol{H}_{\mathcal{D}, \lambda} := (\boldsymbol{\Phi}_{\mathcal{D}}^\top \boldsymbol{\Phi}_{\mathcal{D}} + \lambda I).$$

---

**Algorithm 1** Federated Learning with Newton Method (`FedNewton`)

---

**Input:** Local training data subset $\mathfrak{D}_j$, $\forall j \in [m]$. Feature mapping $\phi : \mathcal{X} \to \mathbb{R}^M$.

**Output:** The global estimator $\bar{w}_{\mathcal{D},\lambda}^T$.

1: **Local machines:** Compute feature mapping $\boldsymbol{\Phi}_{\mathfrak{D}_j}$, $\boldsymbol{H}_{\mathfrak{D}_j,\lambda} = (\boldsymbol{\Phi}_{\mathfrak{D}_j}^\top \boldsymbol{\Phi}_{\mathfrak{D}_j} + \lambda I)$, $\boldsymbol{H}_{\mathfrak{D}_j,\lambda}^{-1}$ and
   $\boldsymbol{\Phi}_{\mathfrak{D}_j}^\top \boldsymbol{y}_{\mathfrak{D}_j}$ for any $j \in [m]$.

2: **Local machines:** Initialize the local estimators by $\boldsymbol{w}_{\mathfrak{D}_j,\lambda}^0 = \boldsymbol{H}_{\mathfrak{D}_j,\lambda}^{-1} \boldsymbol{\Phi}_{\mathfrak{D}_j}^\top \boldsymbol{y}_{\mathfrak{D}_j}$ and upload them
   to the global server ($\uparrow$).

3: **Global server:** Initialize the solution by $\bar{\boldsymbol{w}}_{\mathcal{D},\lambda}^0 = \sum_{j=1}^m p_j \boldsymbol{w}_{\mathfrak{D}_j,\lambda}^0$, and send it to the local nodes
   ($\downarrow$).

4: **for** $t = 1$ to $T$ **do**

5:    **Local machines:** Compute local gradients $\boldsymbol{g}_{\mathfrak{D}_j,\lambda}^{t-1} = \boldsymbol{H}_{\mathfrak{D}_j,\lambda} \bar{\boldsymbol{w}}_{\mathcal{D},\lambda}^{t-1} - \boldsymbol{\Phi}_{\mathfrak{D}_j}^\top \boldsymbol{y}_{\mathfrak{D}_j}$ and upload
      them to global server ($\uparrow$).

6:    **Global server:** Compute the global gradient $\boldsymbol{g}_{\mathfrak{D},\lambda}^{t-1} = \sum_{j=1}^m p_j \boldsymbol{g}_{\mathfrak{D}_j,\lambda}^{t-1}$ and send it to local
      nodes ($\downarrow$).

7:    **Local machines:** Compute the local updates $\boldsymbol{H}_{\mathfrak{D}_j,\lambda}^{-1} \boldsymbol{g}_{\mathfrak{D},\lambda}^{t-1}$ and upload it to the global server
      ($\uparrow$).

8:    **Global server:** Update the global estimator $\bar{\boldsymbol{w}}_{\mathcal{D},\lambda}^t = \bar{\boldsymbol{w}}_{\mathcal{D},\lambda}^{t-1} - \sum_{j=1}^m p_j \boldsymbol{H}_{\mathfrak{D}_j,\lambda}^{-1} \boldsymbol{g}_{\mathfrak{D},\lambda}^{t-1}$ and
      communicate it to local machines ($\downarrow$).

9: **end for**

---

From equation 3, the global gradient $\boldsymbol{g}_{\mathcal{D},\lambda}$ and Hessian $\boldsymbol{H}_{\mathcal{D},\lambda}$ is the key to achieving the centralized model $\boldsymbol{w}_{\mathcal{D},\lambda}$. Note that, since the fact $\boldsymbol{\Phi}_{\mathcal{D}}^\top \boldsymbol{\Phi}_{\mathcal{D}} = \sum_{j=1}^m p_j \boldsymbol{\Phi}_{\mathfrak{D}_j}^\top \boldsymbol{\Phi}_{\mathfrak{D}_j}$ for data partition $\mathcal{D} = \bigcup_{j=1}^m \mathfrak{D}_j$, one can easily obtain the following property for the global gradient and global Hessian.

**Proposition 1** (Partitonability). *If the loss is squared loss, the global gradient and Hessian matrix consist of the local ones, i.e.* $\boldsymbol{g}_{\mathcal{D},\lambda} = \sum_{j=1}^m p_j \boldsymbol{g}_{\mathfrak{D}_j,\lambda}$ *and* $\boldsymbol{H}_{\mathcal{D},\lambda} = \sum_{j=1}^m p_j \boldsymbol{H}_{\mathfrak{D}_j,\lambda}$.

**Remark 1** (Computation of local inverse Hessian). *The compute of the inverse of local Hessians* $\boldsymbol{H}_{\mathfrak{D}_j,\lambda}^{-1}$ *is time consuming* $\mathcal{O}(|\mathfrak{D}_j|M^2 + M^3)$, *which is a common problem in second-order optimization Bottou et al. (2018). There are many classic work to reduce the time complexity of the inverse of Hessian, i.e. BFGS Broyden (1970), L-BFGS Liu & Nocedal (1989), inexact Newton Dembo et al. (1982), Gauss-Newton Schraudolph (2002) and Newton sketch Pilanci & Wainwright (2017). Those techniques can be used to improve the efficiency of* `FedNewton`, *but it is beyond the scope of this paper. We focus on theoretical novelties and leave further computational improvements in the future.*

**Remark 2** (Feature mapping instead of kernel methods). *Without loss of generality, we assume the feature mappings are finite dimensional* $\phi : \mathcal{X} \to \mathbb{R}^M$, *which covers a wide range of generalized linear models, for example neural networks Neal (1995); Jacot et al. (2018), kernel methods Vapnik (2000), random features Rahimi & Recht (2007); Le et al. (2013); Yang et al. (2014), and random sketching Woodruff et al. (2014); Yang et al. (2017).*

## 3  FEDERATED LEARNING WITH NEWTON METHOD

Motivated by recent gradient-based distributed learning Wang et al. (2018); Lin et al. (2020), we propose a Newton-type federated learning method to quantity the impact of data heterogeneity and model heterogeneity. Using Proposition 1, the exact Federated Newton's method communicate local Hessians $\boldsymbol{H}_{\mathfrak{D}_j,\lambda}$ for computing the global Hessian matrix equation 3 whose the communication complexity is $\boldsymbol{O}(M^2)$, which is infeasible in federated learning. To reduce communication costs, we propose `FedNewton` that approximates the Newton's updates with the global gradient and local Hessian matrices, such that

$$\boldsymbol{H}_{\mathcal{D},\lambda}^{-1} \boldsymbol{g}_{\mathcal{D},\lambda} \approx \sum_{j=1}^m p_j \boldsymbol{H}_{\mathfrak{D}_j,\lambda}^{-1} \boldsymbol{g}_{\mathcal{D},\lambda}. \tag{4}$$

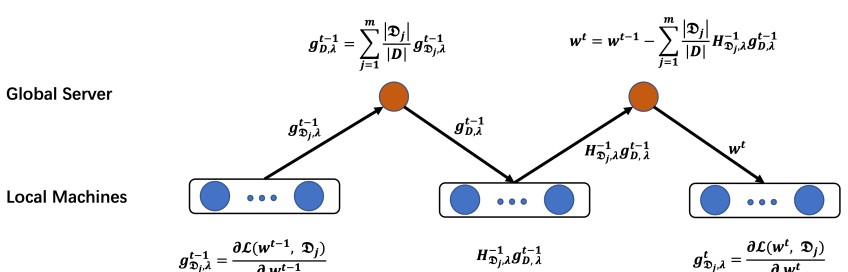

Figure 1: The computations and communications in the $t$-th iteration for `FedNewton`.

The global learner $\bar{f}^t_{\mathcal{D},\lambda}(\boldsymbol{x}) = \langle \bar{\boldsymbol{w}}^t_{\mathcal{D},\lambda}, \phi(\boldsymbol{x}) \rangle$ is updated by

$$\bar{\boldsymbol{w}}^t_{\mathcal{D},\lambda} = \bar{\boldsymbol{w}}^{t-1}_{\mathcal{D},\lambda} - \sum_{j=1}^{m} p_j \boldsymbol{H}^{-1}_{\mathfrak{D}_j,\lambda} \boldsymbol{g}^{t-1}_{\mathcal{D},\lambda}, \tag{5}$$

where $\bar{\boldsymbol{w}}^t_{\mathcal{D},\lambda}$ is the model after $t$ iterations and the global gradient is $\boldsymbol{g}^{t-1}_{\mathcal{D},\lambda} = \sum_{j=1}^{m} p_j \boldsymbol{g}^{t-1}_{\mathfrak{D}_j,\lambda}$ from Proposition 1. The approximation error between equation 3 and equation 5 is analyzed in Section 4. Without loss of generality, we present the details of `FedNewton` in Algorithm 1 and Figure 1, which includes two times communications as the first-order methods in per round. Note that, the algorithm uploads local Newton updates $\boldsymbol{H}^{-1}_{\mathfrak{D}_j,\lambda} \boldsymbol{g}^{t-1}_{\mathcal{D},\lambda} \in \mathbb{R}^M$ instead of local inverse Hessians $\boldsymbol{H}^{-1}_{\mathfrak{D}_j,\lambda} \in \mathbb{R}^{M \times M}$, reducing communication costs from $\boldsymbol{O}(M^2)$ to $\boldsymbol{O}(M)$.

**Computational complexity analysis.** With finite-dimensional feature mappings $\phi : \mathcal{X} \to \mathbb{R}^M$, we compute time complexity, space complexity, and communication complexity of `FedNewton`. The space complexity on the $j$-th local machine is $\mathcal{O}(|\mathfrak{D}_j|M + M^2)$ to store $\boldsymbol{\Phi}_{\mathfrak{D}_j}$, $\boldsymbol{H}_{\mathfrak{D}_j,\lambda}$ and $\boldsymbol{H}^{-1}_{\mathfrak{D}_j,\lambda}$, while the global server requires $\mathcal{O}(mM)$ space to store $\boldsymbol{g}_{\mathfrak{D}_j,\lambda}$ and $\boldsymbol{H}^{-1}_{\mathfrak{D}_j,\lambda} \boldsymbol{g}_{\mathcal{D},\lambda}$. Before the iterations, the computations of $\boldsymbol{H}_{\mathfrak{D}_j,\lambda}$ and $\boldsymbol{H}^{-1}_{\mathfrak{D}_j,\lambda}$ costs $\boldsymbol{O}(|\mathfrak{D}_j|M^2 + M^3)$ time. In each iteration, the local time complexity is $\mathcal{O}(M^2)$ to compute local gradient $\boldsymbol{g}_{\mathfrak{D}_j,\lambda}$ and local Newton update $\boldsymbol{H}^{-1}_{\mathfrak{D}_j,\lambda} \boldsymbol{g}_{\mathfrak{D},\lambda}$, while the time complexity on the global server is $\mathcal{O}(mM)$ to update the global gradient and estimator. Therefore, the total time complexity is $\mathcal{O}\left(\max_{j \in [m]} |\mathfrak{D}_j|M^2 + M^3 + M^2 t + mMt\right)$.

**Remark 3** (Communication burdens). *The per iteration communication costs of the proposed* `FedNewton` *are 2 times as compared to the first-order FL algorithms, e.g. FedAvg and FedProx, but the number of iterations for* `FedNewton` *is much fewer. The total communication complexity is $\mathcal{O}(Mt)$, the same as most first-order Federated algorithms. Notably, from Theorem 1 the iteration complexity is a linear convergence $t = \Omega(\log(1/\epsilon))$ where $\epsilon$ is the federated error, i.e.,* `FedNewton` *converges exponentially to the global estimator equation 2, while first-order federated algorithms requires a large number of communication rounds $t = \Omega(1/\epsilon)$ Su et al. (2021). Therefore,* `FedNewton` *cannot reduce the communication complexity for once communication as communication-efficient FL algorithms Sattler et al. (2019); Reisizadeh et al. (2020); Wu et al. (2022), but it significantly reduces the number of communication rounds, e.g.,* `FedNewton` *with $t \leq 2$ achieves good predictive performance in Section 7.*

**Remark 4** (Beyond the squared loss). *To quantify the impacts from local sample size, data heterogeneity and model heterogeneity, we apply the squared loss for* `FedNewton` *because it admits closed-form solutions and is convenient for the theoretical analysis. Nevertheless, the proposed algorithm* `FedNewton` *is not applies to a broad range of loss functions as long as they are twice differentiable to compute the gradient $\boldsymbol{g}^{t-1}_{\mathfrak{D}_j,\lambda}$ and the Hessian matrix $\boldsymbol{H}_{\mathfrak{D}_j,\lambda}$. If the Hessian is independent from the weights, the compute of local Hessians can be out of the loop, e.g. ReLU and the squared loss. However, if the Hessian is relevant to the weights, for example exponential loss functions and trigonometric loss functions, we should compute the local Hessians for all iterations, causing huge computational burdens. For other type loss functions, the weights can be initialized as $\bar{\boldsymbol{w}}^0_{\mathcal{D},\lambda} = \boldsymbol{0}$.*

## 4 MAIN RESULTS

In this section, to explore the factors that affect performance, we derive the excess risk bounds for `FedNewton` in homogeneous settings and heterogeneous settings, respectively.

### 4.1 NOTATIONS AND ASSUMPTIONS

We consider a broader scenario for federated learning, where the local training sets contain both heterogenous inputs (covariate shift) $\mathfrak{D}_j \sim \rho_j$ and different responses (concept shift) $\boldsymbol{y}_{\mathfrak{D}_j} \sim \rho_j(y|\boldsymbol{x})$. The concept shift is represented as

$$f^*(\boldsymbol{x}) = \int_{\mathcal{Y}} y d\rho(y|\boldsymbol{x}), \ \boldsymbol{x} \in \mathcal{X}, \qquad f_j^*(\boldsymbol{x}) = \int_{\mathcal{Y}} y d\rho_j(y|\boldsymbol{x}), \ \boldsymbol{x} \in \mathcal{X}, \ j \in [m], \qquad (6)$$

where $f_j^*$ is the underlying mechanism governing the true responses on the $j$-th worker. Give a $\boldsymbol{x} \in \mathcal{X}$ and $j, k, \in [m]$, the responses may be different $f_j^*(\boldsymbol{x}) \neq f_k^*(\boldsymbol{x})$ when $j \neq k$.

**Definition 1** (Operators with feature mapping $\phi$). *Using the feature mapping $\phi : \mathcal{X} \to \mathcal{H}_K$, $\forall \boldsymbol{\beta} \in \mathcal{H}_K$, the covariance operators $C, C_j, C_{\mathcal{D}}, C_{\mathfrak{D}_j} : \mathcal{H}_K \to \mathcal{H}_K$ are defined as*

$$C\boldsymbol{\beta} = \int_X \langle \boldsymbol{\beta}, \phi(\boldsymbol{x}) \rangle \phi(\boldsymbol{x}) d\rho_X(\boldsymbol{x}), \qquad C_{\mathcal{D}}\boldsymbol{\beta} = \frac{1}{|\mathcal{D}|} \sum_{i=1}^{|\mathcal{D}|} \langle \boldsymbol{\beta}, \phi(\boldsymbol{x}_i) \rangle \phi(\boldsymbol{x}_i), \ \forall \ (\boldsymbol{x}_i, y_i) \in \mathcal{D},$$

$$C_j\boldsymbol{\beta} = \int_X \langle \boldsymbol{\beta}, \phi(\boldsymbol{x}) \rangle \phi(\boldsymbol{x}) d\rho_j(\boldsymbol{x}), \qquad C_{\mathfrak{D}_j}\boldsymbol{\beta} = \frac{1}{|\mathfrak{D}_j|} \sum_{i=1}^{|\mathfrak{D}_j|} \langle \boldsymbol{\beta}, \phi(\boldsymbol{x}_i) \rangle \phi(\boldsymbol{x}_i), \ \forall \ (\boldsymbol{x}_i, y_i) \in \mathfrak{D}_j.$$

Note that, $C_{\mathcal{D}} = \boldsymbol{\Phi}_{\mathcal{D}}^\top \boldsymbol{\Phi}_{\mathcal{D}}$, $C_{\mathfrak{D}_j} = \boldsymbol{\Phi}_{\mathfrak{D}_j}^\top \boldsymbol{\Phi}_{\mathfrak{D}_j}$ are the empirical covariance operators on $\mathcal{D}$ and $\mathfrak{D}_j$, while $C = \mathbb{E}_\rho[C_{\mathcal{D}}], C_j = \mathbb{E}_{\rho_j}[C_{\mathfrak{D}_j}]$ are their expected counterparts.

For the sake of readability, we provide some notations

$$\mathcal{P}_{\mathfrak{D}_j, \lambda} := \|(C_{\mathfrak{D}_j} + \lambda I)^{-1}(C_j + \lambda I)\|, \qquad \mathcal{R}_{\mathfrak{D}_j, \lambda} := \|(C_j + \lambda)^{-1}(C_j - C_{\mathfrak{D}_j})\|,$$
$$\Delta_{\mathfrak{D}_j} := \|C - C_j\|, \qquad\qquad\qquad \Delta_{f_j} := \|f^* - f_j^*\|.$$

The quantities $\mathcal{P}_{\mathfrak{D}_j, \lambda}$ and $\mathcal{R}_{\mathfrak{D}_j, \lambda}$ measure the similarity between the expected covariance operator and its empirical counterpart. From contraction inequalities for self-adjoint operators, a larger number of local samples $|\mathfrak{D}_j|$ leads to smaller $\mathcal{P}_{\mathfrak{D}_j, \lambda}$ and $\mathcal{R}_{\mathfrak{D}_j, \lambda}$. Note that, $\Delta_{\mathfrak{D}_j}$ measures the data heterogeneity on the expected covariance operator, while $\Delta_{f_j}$ measures the model heterogeneity on the true regressions.

We let $\|f\|_2 = \sqrt{\langle f, f \rangle} = \sqrt{\int_X |f(\boldsymbol{x})|^2 d\mathbb{P}(\boldsymbol{x})}$ denote the $L^2(\mathbb{P})$ norm and $L^2(\mathbb{P}) = \{f : \mathcal{X} \to \mathbb{R} \mid \|f\|_2^2 < \infty\}$. Throughout this paper, we assume the outputs are bounded $|y| \leq B$ almost surely for some $B > 0$ and $\kappa := \|\phi(\boldsymbol{x})\|_K < \infty$ for any $\boldsymbol{x} \in \mathcal{X}$.

**Assumption 1** (Federated capacity condition). *For $\lambda \in (0, 1)$, we define the effective dimensions on the global distribution $\rho$ and local distributions $\rho_j$, $\forall j \in [m]$ as*

$$\mathcal{N}(\lambda) = Tr(C(C + \lambda I)^{-1}), \ \mathcal{N}_j(\lambda) = Tr(C_j(C_j + \lambda I)^{-1}).$$

*Assume there exists $Q > 0$ and $\gamma \in [0, 1]$, such that*

$$\max\left(\mathcal{N}(\lambda), \mathcal{N}_1(\lambda), \cdots, \mathcal{N}_m(\lambda)\right) \leq Q^2 \lambda^{-\gamma}.$$

**Assumption 2** (Source condition). *Define the integral operators $L : L^2(\mathbb{P}) \to L^2(\mathbb{P})$,*

$$(Lg)(\cdot) = \int_X \langle \phi(\cdot), \phi(\boldsymbol{x}) \rangle g(\boldsymbol{x}) d\rho_X(\boldsymbol{x}), \quad \forall \ g \in L^2(\mathbb{P}).$$

*Assume there exists $R > 0$, $r > 0$, such that $\|L^{-r}f^*\| \leq R$. where the operator $L^r$ denotes the $r$-th power of $L$ as a compact and positive operator.*

Capacity condition and source condition are standard assumptions in the optimal statistical learning for the KRR related literature Caponnetto & De Vito (2007); Smale & Zhou (2007); Rudi & Rosasco (2017); Lin & Cevher (2020); Liu et al. (2021). The effective dimensions $\mathcal{N}(\lambda)$ and $\mathcal{N}_j(\lambda)$ measure the capacities of the RKHS $\mathcal{H}_K$ on the global distribution $\rho$ and the local distributions $\rho_j$, $\forall j \in [m]$.

Here, we modify the conventional capacity condition for federated learning to impose constraints on local estimators. Note that, for effective dimensions, it holds $1/2 \leq \max(\mathcal{N}(\lambda), \mathcal{N}_1(\lambda), \cdots, \mathcal{N}_m(\lambda)) \leq \kappa^2 \lambda^{-1}$ Rudi et al. (2015). Assumption 1 reflects the variance of the estimator. A larger $\gamma$ leads to a larger $\mathcal{H}_K$ and $\gamma = 1$ corresponds to the capacity independence case. Assumption 2 controls the bias of an estimator, which reflects the regularity of the estimator. The bigger $r$ leads to the stronger regularity of the regression and the easier learning problem. The general settings ($r = 1/2, \gamma = 1$) lead to $\boldsymbol{O}(1/\sqrt{|D|})$ convergence rates for KRR related approaches.

## 4.2 ERROR DECOMPOSITION

**Theorem 1.** *Let $f_{\mathcal{D},\lambda}, \bar{f}_{\mathcal{D},\lambda}^t, f^*$ be defined according to equation 2, equation 5 and equation 6. Then, the following error decomposition holds*

$$\|\bar{f}_{\mathcal{D},\lambda}^t - f^*\| \leq \underbrace{\|\bar{f}_{\mathcal{D},\lambda}^t - f_{\mathcal{D},\lambda}\|}_{\text{federated error}} + \underbrace{\|f_{\mathcal{D},\lambda} - f^*\|}_{\text{centralized excess risk}}, \tag{7}$$

*and the federated error for* FedNewton *is bounded by:*

$$\|\bar{f}_{\mathcal{D},\lambda}^t - f_{\mathcal{D},\lambda}\|_2 \leq \Upsilon^t \left\| (C + \lambda I)^{1/2} (\bar{\boldsymbol{w}}_{\mathcal{D},\lambda}^0 - \boldsymbol{w}_{\mathcal{D},\lambda}) \right\|_K,$$

*where $\Upsilon = \sum_{j=1}^m p_j \mathcal{P}_{\mathfrak{D}_j,\lambda} \left( 2\mathcal{R}_{\mathfrak{D}_j,\lambda} + \frac{\Delta_{\mathfrak{D}_j}}{\lambda} \right) \left( 1 + \frac{\Delta_{\mathfrak{D}_j}}{\lambda} \right).$*

In the above theorem, we decompose the excess risk for FedNewton into two parts: the federated error $\|\bar{f}_{\mathcal{D},\lambda}^t - f_{\mathcal{D},\lambda}\|$ and the excess risk for the centralized KRR $\|f_{\mathcal{D},\lambda} - f^*\|$. Since the generalization analysis for $\|f_{\mathcal{D},\lambda} - f^*\|$ is standard Caponnetto & De Vito (2007); Smale & Zhou (2007), we focus on the federated error $\|\bar{f}_{\mathcal{D},\lambda}^t - f_{\mathcal{D},\lambda}\|$.

From Theorem 1, we find that the value of $\Upsilon$ determines the effectiveness of multiple iterations. If $\Upsilon \geq 1$, FedNewton with multiple communications is worse than oneshot federated learning (DKRR). However, when $\Upsilon < 1$, the federated error decreases exponentially and the rate of convergence is referred to as *linear convergence* in the optimization literature Bottou et al. (2018). The quantities $\mathcal{P}_{\mathfrak{D}_j,\lambda}$ and $\mathcal{R}_{\mathfrak{D}_j,\lambda}$ measure the similarity between $C_{\mathfrak{D}_j}$ and $C_j$ where those quantities decrease as the local sample size $|\mathfrak{D}_j|$ increases. Because $\Upsilon$ is proportional to $\mathcal{P}_{\mathfrak{D}_j,\lambda}$, $\mathcal{P}_{\mathfrak{D}_j,\lambda}$ and $\Delta_{\mathfrak{D}_j}$, the *linear convergence* requires both a sufficient number of local examples $|\mathfrak{D}_j|$ and moderate data heterogeneity $\Delta_{\mathfrak{D}_j}$. If $t = 0$, the above error bound degrades into that for DKRR $\|\bar{f}_{\mathcal{D},\lambda} - f_{\mathcal{D},\lambda}\|_2 \leq \left\| (C + \lambda I)^{1/2} (\bar{\boldsymbol{w}}_{\mathcal{D},\lambda}^0 - \boldsymbol{w}_{\mathcal{D},\lambda}) \right\|_K.$

**Theorem 2.** *Under Assumption 2, with a high probability $1 - \delta$, $\forall \delta \in (0,1)$, the federated error can be bounded*

$$\|\bar{f}_{\mathcal{D},\lambda}^t - f_{\mathcal{D},\lambda}\|_2 \lesssim \Upsilon^t \sum_{j=1}^m p_j \sqrt{1 + \frac{\Delta_{\mathfrak{D}_j}}{\lambda}} \left( 2\mathcal{R}_{\mathfrak{D}_j,\lambda} + \frac{(1 + \mathcal{R}_{\mathfrak{D}_j,\lambda})\Delta_{\mathfrak{D}_j}}{\lambda} \right) \cdot$$

$$\left( \left( \frac{1}{|\mathfrak{D}_j|\sqrt{\lambda}} + \sqrt{\frac{\mathcal{N}(\lambda)}{|\mathfrak{D}_j|}} \right) \log \frac{2}{\delta} + \frac{\Delta_{\mathfrak{D}_j}}{\lambda} + \Delta_{f_j} \right).$$

Theorem 2 illustrates the key factors that affect the federated error: the discrepancy between expected and empirical covariance operators $\mathcal{R}_{\mathfrak{D}_j,\lambda}$, the covariate shift $\Delta_{\mathfrak{D}_j}$, and the model heterogeneity $\Delta_{f_j}$. The smaller these factors, the smaller the federated error. The federated error results from three parts: distributed error $\frac{1}{\sqrt{\lambda}|\mathfrak{D}_j|} + \sqrt{\frac{\mathcal{N}(\lambda)}{|\mathfrak{D}_j|}}$, covariate shift $\Delta_{\mathfrak{D}_j}/\lambda$ and concept shift $\Delta_{f_j}$. Specifically, as the increase of local sample size, the distributed error decreases. However, the concept shifts $\Delta_{f_j}$ is a constant and it will dominate the federated error when model heterogeneity $\Delta_{f_j}$ is large. In the case $\Upsilon < 1$, iterators can reduce the federated error, alleviating the entire federated error term.

### 4.3 Homogeneous Setting

**Theorem 3.** *Let $\delta \in (0, 1/3]$, $\lambda = |\mathcal{D}|^{\frac{-1}{2r+\gamma}}$ and $2r + \gamma \geq 1$. Under Assumptions 1, 2, if $\Delta_{\mathfrak{D}_j} = 0$ and $\Delta_{f_j} = 0$, with the probability at least $1 - 3\delta$, it holds*

$$\|\bar{f}_{\mathcal{D},\lambda}^t - f^*\|_2 \lesssim \Upsilon^t \sum_{j=1}^m p_j \aleph_j \log^2 \frac{2}{\delta} + |\mathcal{D}|^{\frac{-r}{2r+\gamma}} \log \frac{2}{\delta}.$$

*Here, $\aleph_j$ and $\Upsilon$ have different values w.r.t local sample size*

$$\aleph_j = \begin{cases} |\mathfrak{D}_j|^{-2} |\mathcal{D}|^{\frac{1.5}{2r+\gamma}}, & \text{if } |\mathfrak{D}_j| \lesssim |\mathcal{D}|^{\frac{1-\gamma}{2r+\gamma}} \\ |\mathfrak{D}_j|^{-1.5} |\mathcal{D}|^{\frac{1+0.5\gamma}{2r+\gamma}}, & \text{if } |\mathcal{D}|^{\frac{1-\gamma}{2r+\gamma}} \lesssim |\mathfrak{D}_j| \lesssim |\mathcal{D}|^{\frac{1}{2r+\gamma}} \\ |\mathfrak{D}_j|^{-1} |\mathcal{D}|^{\frac{1+\gamma}{4r+2\gamma}}, & \text{if } |\mathcal{D}|^{\frac{1}{2r+\gamma}} \lesssim |\mathfrak{D}_j| \lesssim |\mathcal{D}|^{\frac{2r+\gamma+1}{4r+2\gamma}} \\ |\mathcal{D}|^{\frac{-r}{2r+\gamma}}, & \text{if } |\mathfrak{D}_j| \gtrsim |\mathcal{D}|^{\frac{2r+\gamma+1}{4r+2\gamma}}, \end{cases}$$

*and $\Upsilon = 2 \sum_{j=1}^m p_j \mathcal{P}_{\mathfrak{D}_j,\lambda} \mathcal{R}_{\mathfrak{D}_j,\lambda}$ holds*

$$\begin{cases} \Upsilon \geq 1, & \text{if } |\mathfrak{D}_j| \lesssim |\mathcal{D}|^{\frac{1}{2r+\gamma}} \\ \Upsilon \lesssim \frac{|\mathcal{D}|^{\frac{1}{2r+\gamma}}}{|\mathfrak{D}_j|} < 1, & \text{otherwise.} \end{cases}$$

Note that, the second term in the above bound is from the centralized model $\|f_{\mathcal{D},\lambda} - f^*\|_2$, where the learning rate $\boldsymbol{O}(|\mathcal{D}|^{\frac{-r}{2r+\gamma}})$ is optimal in a minimax sense Caponnetto & De Vito (2007). The performance of `FedNewton` in the homogeneous setting is only affected by the local sample size. We discuss the above result in three parts. First, when the number of local examples is limited $|\mathfrak{D}_j| \lesssim |\mathcal{D}|^{\frac{1}{2r+\gamma}}$, in another word the number of local machines is larger than $m \gtrsim |\mathcal{D}|^{\frac{2r+\gamma-1}{2r+\gamma}}$, the federated error dominates the excess risk and fails to achieve the optimal rate, where the convergence rates are slower than $\mathcal{O}(|\mathcal{D}|^{\frac{\gamma-1}{4r+2\gamma}})$. Meanwhile, when the number of local examples is limited, it leads to $\Upsilon \geq 1$ and multiple communications hurt the performance. Second, when $|\mathcal{D}|^{\frac{1}{2r+\gamma}} \lesssim |\mathfrak{D}_j| \lesssim |\mathcal{D}|^{\frac{2r+\gamma+1}{4r+2\gamma}}$, although the convergence rates of federated error are still not the optimal, the iterator $\Upsilon$ is smaller than one, leading to a linear convergence. As the increase of communications $t \to \infty$, the centralized excess risk will dominate the error bound that achieves the optimal rate. Third, with a large number of local examples $|\mathfrak{D}_j| \gtrsim |\mathcal{D}|^{\frac{2r+\gamma+1}{4r+2\gamma}}$, even with insufficient communications $t \to 0$, the error bound still achieves the optimal rate $\boldsymbol{O}(|\mathcal{D}|^{\frac{-r}{2r+\gamma}})$.

Theorem 3 can be further simplified in some special cases. For example, we consider the general case $(r = 1/2, \gamma = 1)$, where $r = 1/2$ is equivalent to assuming $f^* \in \mathcal{H}_K$ and $\gamma = 1$ is the capacity independent case. The learning rate achieves $\boldsymbol{O}(1/\sqrt{|\mathcal{D}|})$ when $|\mathfrak{D}_j| \gtrsim |\mathcal{D}|^{0.5}$ with multiple iterations or $|\mathfrak{D}_j| \gtrsim |\mathcal{D}|^{0.75}$ with only one communication.

**Remark 5.** *The existing theoretical guarantees for DKRR Zhang et al. (2015); Guo et al. (2017); Lin & Cevher (2020) focused on how to achieve the optimal rate by a sufficient number of local examples (or lower the number of partitions), but they ignored the sub-optimal case that the local sample size is fixed and insufficient. However, in federated learning, the number of partitions is fixed and local examples are generated locally, such that sub-optimal cases are more general. Theorem 3 illustrate that a sufficient number of local examples is crucial for both learning rates (in generalization) and convergence rate (in optimization).*

**Remark 6** (Finite dimensional case). *In the proofs of theoretical findings, we consider the estimator in RKHS with $\boldsymbol{w} \in \mathcal{H}_K$. However, the finite-dimensional cases are more general, i.e. $\boldsymbol{w} \in \mathbb{R}^M$ in Algorithm 1, where the feature mappings are explicit and can be neural networks or random features Rahimi & Recht (2007). With a simple modification of our proofs, one can derive similar results for finite-dimensional cases. In particular, under same assumptions of Theorem 3 and $(r = 1/2, \gamma = 0)$, then with high probability, $\|\bar{f}_{\mathcal{D},\lambda}^t - f^*\|_2 \lesssim |\mathfrak{D}_j|^{-2} |\mathcal{D}|^{1.5} + \sqrt{M/|\mathcal{D}|}$, provided that $|\mathcal{D}| \gtrsim M \log M$.*

*As shown in Rudi & Rosasco (2017), a large number of random features $M \gtrsim |\mathcal{D}|^{\frac{1+\gamma(2r-1)}{2r+\gamma}}$ can guarantee the optimal rates for $\|f_{\mathcal{D},\lambda} - f^*\|_2$, and thus we can also provide similar results as Theorem 3.*

### 4.4 HETEROGENEOUS SETTING

**Theorem 4.** *Let $\delta \in (0, 1/3]$, $\lambda = |\mathcal{D}|^{\frac{-1}{2r+\gamma}}$ and $2r + \gamma \geq 1$. Under Assumptions 1, 2, with the probability at least $1 - 3\delta$, the excess risk bound for* `FedNewton` *holds*

$$\|\bar{f}^t_{\mathcal{D},\lambda} - f^*\|_2 \lesssim \Upsilon^t \sum_{j=1}^{m} p_j \sqrt{1 + \frac{\Delta_{\mathfrak{D}_j}}{\lambda}} (\aleph_j + \Pi_j) \log^2 \frac{2}{\delta} + |\mathcal{D}|^{\frac{-r}{2r+\gamma}} \log \frac{2}{\delta}.$$

*Here, $\Upsilon = \sum_{j=1}^{m} p_j \mathcal{P}_{\mathfrak{D}_j,\lambda}(2\mathcal{R}_{\mathfrak{D}_j,\lambda} + \frac{\Delta_{\mathfrak{D}_j}}{\lambda})(1 + \frac{\Delta_{\mathfrak{D}_j}}{\lambda})$, $\aleph_j$ is same to Theorem 3 and*

$$\Pi_j = \begin{cases} \frac{|\mathcal{D}|^{\frac{2}{2r+\gamma}}}{|\mathfrak{D}_j|} \Delta_{\mathfrak{D}_j} + \frac{|\mathcal{D}|^{\frac{1}{2r+\gamma}}}{|\mathfrak{D}_j|} \Delta_{f_j}, & \text{if } |\mathfrak{D}_j| \lesssim |\mathcal{D}|^{\frac{1}{2r+\gamma}} \\ (1 + |\mathcal{D}|^{\frac{1}{2r+\gamma}} \Delta_{\mathfrak{D}_j})(\Delta_{f_j} + |\mathcal{D}|^{\frac{1}{2r+\gamma}} \Delta_{\mathfrak{D}_j}), & \text{otherwise.} \end{cases}$$

We add some comments on the above theorem. First, when the local sample size is insufficient $|\mathfrak{D}_j| \lesssim |\mathcal{D}|^{\frac{1}{2r+\gamma}}$ or the data heterogeneity is considerable, we have $\Upsilon \geq 1$, and communications hurt the performance. Meanwhile, since the federated error $\sqrt{1 + \Delta_{\mathfrak{D}_j}/\lambda}(\aleph_j + \Pi_j)$ depends on $|\mathfrak{D}_j|, \Delta_{\mathfrak{D}_j}$, and $\Delta_{f_j}$, the learning rate is far from the optimal rate. Second, when the number of local examples is sufficient $|\mathfrak{D}_j| \gtrsim |\mathcal{D}|^{\frac{1}{2r+\gamma}}$ and data heterogeneity is small, it holds $\Upsilon < 1$ where communications can improve the generalization ability of `FedNewton`. In this case, the federated error $\|\bar{f}^t_{\mathcal{D},\lambda} - f_{\mathcal{D},\lambda}\|$ converge exponentially fast. If $t$ is large enough, the error bound in Theorem 4 depends on the centralized excess risk $\|f_{\mathcal{D},\lambda} - f^*\|_2$ and achieves the optimal learning rate.

The learning rate of generalization bound in Theorem 4 is determined by four factors: the local sample size $|\mathfrak{D}_j|$, the covariate shift $\Delta_{\mathfrak{D}_j}$, the response shift $\Delta_{f_j}$ and the number of iterations $t$. Furthermore, the iterator value $\Upsilon$ depends on $|\mathfrak{D}_j|$ and $\Delta_{\mathfrak{D}_j}$, such that these two values are important factors for both fast convergences (in optimization) and the learning rates (in generalization).

**Remark 7** (How to achieve the optimal rate in federated learning?). *The value of $\Upsilon < 1$ is key to obtaining a linear convergence rate and the optimal learning rate, where it depends on both local sample sizes $\Upsilon \propto \mathcal{R}_{\mathfrak{D}_j,\lambda} \propto |\mathfrak{D}_j|$ and data heterogeneity $\Upsilon \propto \Delta_{\mathfrak{D}_j}$. Note that, $\Delta_{\mathfrak{D}_j}$ measures the intrinsic discrepancy between local distributions and the global one, and thus it is a fixed value independent from the local sample size. Therefore, since $\Delta_{\mathfrak{D}_j}$ is a constant, we can obtain $\Upsilon < 1$ with a large number of local examples generated by local machines. And then, with a large number of iterations when $\Upsilon < 1$, the federated error, depending on both data heterogeneity and model heterogeneity, can become small enough to be negligible. In this case, a large number of local examples can guarantee both a linear convergence rate (for federated error) and the optimal learning rate (from the centralized excess risk). A large number of local examples benefit both optimization and generalization, rather than making tradeoffs between them.*

## 5 COMPARED WITH RELATED WORK

We compare `FedNewton` with recent Newton-type methods, DKRR methods, and first-order FL algorithms in both algorithmic and theoretical fronts. Table 1 reports the main factors that affect the performance, the computational and generalization properties of related work.

**Compared with Newton-type FL methods.** Local Newton-type FL algorithms Yang et al. (2019); Ghosh et al. (2020); Gupta et al. (2021) conducted Newton updates instead of SGD in local machines, which only utilized local information (local SGD & local Hessian). Recent studies Safaryan et al. (2022); Qian et al. (2022) tried to use global information (global SGD & global Hessian) by communicating local Hessian shifts, but it leads to high communication costs $\boldsymbol{O}(M^2)$ per communication. Nevertheless, this work employs mixed information (global SGD & local Hessian) that reduce the communication cost to $\boldsymbol{O}(M)$. More importantly, the existing Newton-type FL work only provided the convergence analysis (optimization) Ghosh et al. (2020); Safaryan et al. (2022); Qian et al. (2022) without out-sample (generalization) error bounds, while this work bridges the optimization and generalization for `FedNewton`, which essentially guarantees its fast convergence and good generalization ability.

**Compared with DKRR.** The time complexities of DKRR approaches solved in kernel space Zhang et al. (2015); Guo et al. (2017) are much higher than that of stochastic optimization methods solved

Table 1: Summary of computational and generalization properties for related work.

| Related Work | $|\mathfrak{D}_j|$ | $\Delta_{\mathfrak{D}_j}$ | $\Delta_{f_j}$ | Training Time | Testing Time | Communication | Conditions | Local Size $|\mathfrak{D}_j|$ | Iteration $t$ | Upper Bound |
|---|---|---|---|---|---|---|---|---|---|---|
| DKRR Zhang et al. (2015) | √ | × | × | $|\mathfrak{D}_j|^3$ | $|\mathcal{D}_{\text{test}}||\mathcal{D}|$ | $|\mathcal{D}|$ | Specific kernels | $\Omega(r^2\kappa^4\log|\mathcal{D}|)$ | $O(1)$ | $O\left(\frac{1}{|\mathcal{D}|}\right)$ |
| DKRR Guo et al. (2017) | √ | × | × | $|\mathfrak{D}_j|^3$ | $|\mathcal{D}_{\text{test}}||\mathcal{D}|$ | $|\mathcal{D}|$ | $r \in [1/2,1]$ | $\Omega(|\mathcal{D}|^{\frac{1+\gamma}{2r+\gamma}})$ | $O(1)$ | $O(|\mathcal{D}|^{\frac{-r}{2r+\gamma}})$ |
| DKRR-SGD Lin & Cevher (2018) | √ | × | × | $|\mathcal{D}|t$ | $|\mathcal{D}_{\text{test}}||\mathcal{D}|$ | $|\mathcal{D}|$ | $r \in [1/2,1]$ | $\Omega(|\mathcal{D}|^{\frac{1}{2r+\gamma}})$ | $O(|\mathcal{D}|^{\frac{2-\gamma}{2r+\gamma}})$ | $O\left(|\mathcal{D}|^{\frac{-r}{2r+\gamma}}\right)$ |
| DKRR-CM Lin et al. (2020) | √ | × | × | $|\mathfrak{D}_j|^3 + |\mathcal{D}||\mathfrak{D}_j|t$ | $|\mathcal{D}_{\text{test}}||\mathcal{D}|$ | $|\mathcal{D}|t$ | $r \in [1/2,1]$ | $\Omega(|\mathcal{D}|^{\frac{2r+\gamma+1}{4r+2\gamma}})$ | $O(\log\frac{1}{\epsilon})$ | $O\left(|\mathcal{D}|^{\frac{-r}{2r+\gamma}}\right)$ |
| FedAvg Su et al. (2021) | × | × | √ | $|\mathfrak{D}_j|M^2 + M^2t + mMt$ | $|\mathcal{D}_{\text{test}}|M$ | $Mt$ | Specific kernels | / | $O(\frac{1}{\eta t})$ | $O\left(\frac{1}{\eta t} + \frac{\Delta_f^2}{|\mathcal{D}|}\right)$ |
| FedProx Su et al. (2021) | × | × | √ | $|\mathfrak{D}_j|M^2 + M^3 + M^2t + mMt$ | $|\mathcal{D}_{\text{test}}|M$ | $Mt$ | Specific kernels | / | $O(\frac{1}{\epsilon})$ | $O\left(\frac{1}{\eta t} + \frac{\Delta_f^2}{|\mathcal{D}|}\right)$ |
| Theorem 3 | √ | × | × | $|\mathfrak{D}_j|M^2 + M^3 + M^2t + mMt$ | $|\mathcal{D}_{\text{test}}|M$ | $Mt$ | $r > 0, 2r + \gamma \geq 1$ | $\Omega(|\mathcal{D}|^{\frac{1}{2r+\gamma}})$ | $O(\log\frac{1}{\epsilon})$ | Theorem 3 |
| Theorem 4 | √ | √ | √ | $|\mathfrak{D}_j|M^2 + M^3 + M^2t + mMt$ | $|\mathcal{D}_{\text{test}}|M$ | $Mt$ | $r > 0, 2r + \gamma \geq 1$ | $\Omega(|\mathcal{D}|^{\frac{1}{2r+\gamma}})$ | $O(\log\frac{1}{\epsilon})$ | Theorem 4 |

Note: The computational complexities are computed in terms of regularized least squared loss. We estimate the upper bounds for $\|f - f^*\|_2 \ \forall f \in L^2(\mathbb{P})$. We denote $\mathcal{D}_{\text{test}}$ the testing data, $\eta$ the step-size for SGD approaches, $\epsilon$ the federated error and $\Delta_f^2 = \sum_{j=1}^m p_j \Delta_{f_j}^2$. For Rademacher complexities based bounds Zhang et al. (2015); Su et al. (2021), specific kernels include kernels with finite-rank or polynomial eigenvalues decay. Integral operator based bounds Guo et al. (2017); Lin & Cevher (2018); Lin et al. (2020) also assume $\gamma \in [0,1]$. We compute exact local solution for FedProx.

in feature space. Both our work and Guo et al. (2017); Lin & Cevher (2018); Lin et al. (2020) are based on integral operator techniques, but DKRR literature assumes all local datasets are drawn i.i.d. from an identical distribution, ignoring the data heterogeneity and model heterogeneity, which makes the proofs much easier than ours. We emphasize the difference between this work and DKRR theories as bellow: 1) DKRR work required a strict condition $r \in [1/2,1]$, while we relax the condition to $r > 0, 2r + \gamma \geq 1$. 2) This work pertains to NonIID data, covering both covariate shift $\Delta_{\mathfrak{D}_j}$ and response shift $\Delta_{f_j}$, DKRR only applied to IID data that is a special case in the homogenous setting $\Delta_{\mathfrak{D}_j} = \Delta_{f_j} = 0$ in Theorem 3. 3) Because of the existence of data heterogeneity and model heterogeneity, we cannot directly estimate the difference between local estimators and global ones, and thus we introduce novel error decompositions for the federated error. 4) This work explores the excess risk bounds in terms of different local sample size ($\aleph_j$ in Theorem 3), covering both optimal and sub-optimal rates, while DKRR work only studied the optimal learning rates with the restrict on the number of partitions, i.e. $m = O(|\mathcal{D}|^{\frac{(2r+\gamma-1)(t+1)}{(2r+\gamma)(t+2)}})$ Lin et al. (2020).

**Compared with first-order methods.** Using the random matrix theory and the local Rademacher complexity, Su et al. (2021) provided the optimal guarantees $\|f - f^*\|_2^2 = O(1/|\mathcal{D}|)$. However, as shown in Theorem 2 Su et al. (2021), it directly assumed all inputs are sampled i.i.d from an identical distribution, ignore the local sample size and the data heterogeneity, while our theoretical results illustrate both the local sample size and the data heterogeneity are crucial to federated learning. Su et al. (2021) also imposed several strict assumptions: 1) the ideal model belongs to the hypothesis space, corresponding to $r \in [1/2,1]$; 2) small hypothesis space with local Rademacher complexity, corresponding $\gamma \to 0$ in our work; 3) specific kernels maybe not suitable to the federated learning tasks and lead to sub-optimal rates. In this work, we remove these three conditions based on the integral operator approach, which makes our theoretical findings applicable to broader settings. Our results illustrate that only a few iterations can guarantee the optimal rates $O(|\mathcal{D}|^{\frac{-2r}{2r+\gamma}})$ when the number of local examples is sufficient and data heterogeneity is moderate, where the convergence rate of federated error is *linear*, while in Su et al. (2021) the learning rate is always affected by model heterogeneity $O(\frac{\sum_{j=1}^m p_j \Delta_{f_j}^2}{|\mathcal{D}|})$ and the convergence rate is *sublinear*.

## 6 CONCLUSION AND FUTURE WORK

In this paper, we present an efficient second-order optimization method for FL. We derive generalization bounds with the optimal rates, which quantify the impacts of local sample size, the data heterogeneity, and the model heterogeneity. In benign cases, the federated error convergence exponentially fast, and thus communications can be small. Our theoretical findings fill the gap between optimization and generalization for federated learning, rather than focusing on one of them. Overall, the techniques presented here highlight new ways for designing efficient algorithms and analyzing both generalization and optimization for FL.

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
