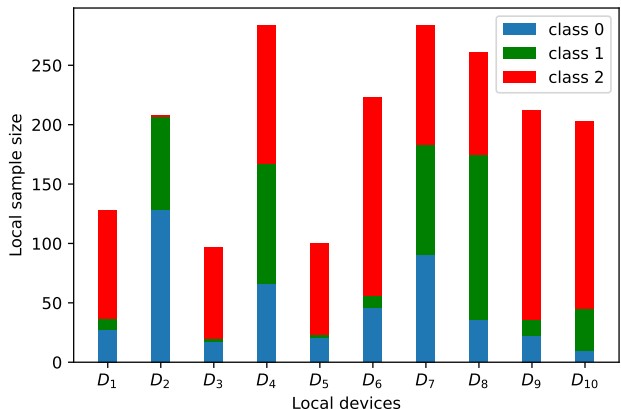

Figure 2: Data partitions for the dna dataset.

## 7 EXPERIMENTS

In this section, we first carry out simulations to corroborate our theoretical statements. Then, we compare the performance of `FedNewton` with related baselines on real-world datasets.

### DATASETS

**1) Synthetic dataset.** Although the existing work Li et al. (2020a); Lin et al. (2020); Su et al. (2021) provide strategies to generate synthetic datasets, these datasets either fail to impose both data heterogeneity and model heterogeneity among devices, or just fit a simple linear problem. Here, we focus on a nonlinear problem $f^*(\boldsymbol{x}) = \min(-\mathbf{1}^\top \boldsymbol{x}, \mathbf{1}^\top \boldsymbol{x})$ with $\boldsymbol{x} \sim \mathcal{N}(0, \mathbf{I})$. On the $j$-th local machine, we generate $\mathfrak{D}_j = (\boldsymbol{X}_j, \boldsymbol{y}_j)$ based on $y = \min(-\boldsymbol{w}^\top \boldsymbol{x}, \boldsymbol{w}^\top \boldsymbol{x}) + \epsilon$, where $\epsilon \sim \mathcal{N}(0, 0.2)$ is the label noise, $\boldsymbol{x}_j \sim \mathcal{N}(\boldsymbol{u}_j, \mathbf{I}), \boldsymbol{u}_j \sim \mathcal{N}(0, \alpha)$ and $\boldsymbol{w}_j \sim \mathcal{N}(\mathbf{1}, \boldsymbol{v}_j), \boldsymbol{v}_j \sim \mathcal{N}(0, \beta)$. Notably, $\alpha$ and $\beta$ control the data heterogeneity and model heterogeneity, respectively. Data heterogeneity and model heterogeneity increase as $\alpha$ and $\beta$ become larger, and the homogeneous setting corresponds to $\alpha = \beta = 0$. We set $d = 10$ and generate $|\mathcal{D}| = 10000$ samples for training, 2500 samples for testing.

**2) Real-world datasets.** We evaluate the compared algorithms on publicly available datasets from LIBSVM Data [1], which provide both training and testing data. To construct a heterogeneous and unbalanced setting, we split these datasets across 10 clients using a Dirichlet distribution $\text{Dir}_K(c)$ Wang et al. (2020), where $c$ is some constant relevant to the level of heterogeneity and unbalanced distribution. For example, the data partition for the *dna* dataset with $\text{Dir}_K(1)$ is reported in Figure 2 where the local datasets are both heterogeneous and unbalanced, which is common in federated learning scenarios.

### EXPERIMENTAL SETTINGS

We compared the proposed `FedNewton` with the baseline (KRR on entire data), DKRR (`FedNewton` with $t = 0$), FedAvg McMahan et al. (2017) and FedProx Li et al. (2020a) with the squared loss equation 1. The estimator can be expressed as $f(\boldsymbol{x}) = \langle \boldsymbol{w}, \phi(\boldsymbol{x}) \rangle$, where $\phi(\boldsymbol{x})$ denotes the feature mapping function. Here, we use random Fourier feature $\phi(\boldsymbol{x}) = 1/\sqrt{M} \cos(\boldsymbol{\Omega}^\top \boldsymbol{x} + \boldsymbol{b})$, where $\phi : \mathbb{R}^d \to \mathbb{R}^M, \boldsymbol{\Omega} \in \mathbb{R}^{d \times M}, \boldsymbol{b} \in \mathbb{R}^M$ and $\boldsymbol{\Omega} \sim \mathcal{N}(0, 1/\sigma^2), \boldsymbol{b} \in U[0, 2\pi]$. We set $M = 200$ for synthetic dataset and $M = 2000$ for real-world datasets. We implement all code based Pytorch and tune the hyperparameters over $\sigma^2 \in \{0.01, 0.1, \cdots, 1000\}$ and $\lambda = \{0.1, 0.01, \cdots, 10^{-7}\}$ by grid search. We report the data statistics and parameter setting in Table 2. All experiments are recorded by averaging results after 10 trials.

---

[1] Available at `https://www.csie.ntu.edu.tw/~cjlin/libsvmtools/`

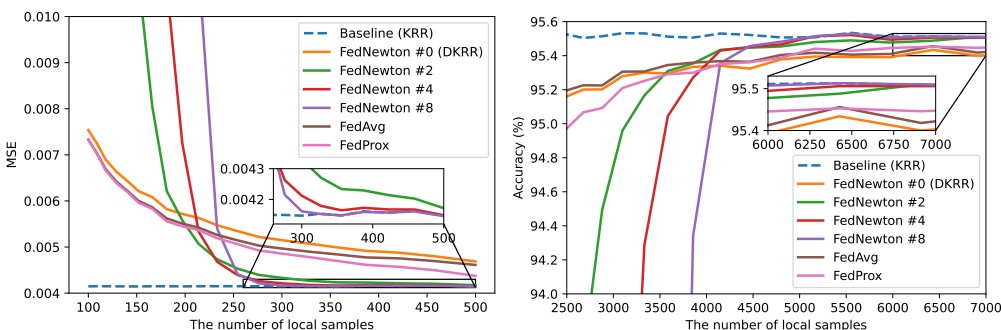

Figure 3: Impact of the number of local samples (left) on the synthetic dataset and MNIST dataset (right). The number of total training samples is fixed, $|\mathfrak{D}_j| = |\mathcal{D}|/m$ and $\Delta_{\mathfrak{D}_j} = \Delta_{f_j} = 0$. The blue dotted line denotes the exact KRR on all training data.

We initialize all iterative methods, including FedAvg, FedProx and `FedNewton`, by $\boldsymbol{w}^0_{\mathfrak{D}_j,\lambda} = \boldsymbol{H}^{-1}_{\mathfrak{D}_j,\lambda} \boldsymbol{\Phi}^\top_{\mathfrak{D}_j} \boldsymbol{y}_{\mathfrak{D}_j}$ rather than $\boldsymbol{w}^0_{\mathfrak{D}_j,\lambda} = \boldsymbol{0}$. DKRR directly averages the initialized models. FedAvg updates local models with $s = 2$ iterations on all local data in each epoch. In Section 7, we estimate the impact of local sample size, data heterogeneity without comparing FedAvg and FedProx. Here, we provide the full comparison with FedAvg and FedProx w.r.t. local sample size and data heterogeneity.

### 7.1 Empirical Validations

We verify the theoretical findings in theorems by exploring how the factors empirically affect the performance on a synthetic dataset that can capture both data heterogeneity and model heterogeneity and the MNIST dataset.

**Impact of local sample size.** We explore the influence of local sample size $|\mathfrak{D}_j|$ by fixing the total sample size $|\mathcal{D}| = 10000$ while varying the number $m$ of local machines, where $|\mathfrak{D}_j| = \frac{|\mathcal{D}|}{m}$. As shown in the first two in Figure 3, when the number of local samples is small, i.e. $|\mathfrak{D}_j| < 200$ for the synthetic dataset and $|\mathfrak{D}_j| < 3300$ for MNIST, `FedNewton` with multiple communications hurts the generalization performance, and more communications lead to worse accuracy, corresponding to the cases $\Upsilon^t > 1$ in Theorem 3. When the local sample size is larger than a threshold, i.e. $|\mathfrak{D}_j| \approx 260$ for the synthetic dataset and $|\mathfrak{D}_j| \approx 4400$ for MNIST, more communications can significantly improve the predictive performance and get closer to the exact KRR, which coincides with the cases $\Upsilon^t < 1$ in Theorem 3. Note that, even with a large number of local examples, there still is a great gap between DKRR and KRR, while `FedNewton` achieves a good approximation to KRR. Meanwhile, both larger $|\mathfrak{D}_j|$ and larger $t$ can improve the approximation ability, validating the theoretical results. Compared to first-order methods, when the local sample size is large enough, `FedNewton` outperforms FedAvg and FedProx. However, `FedNewton` is more sensitive to the number of local examples, and we find that the predictive error explodes when local sample size is small.

**Impact of heterogeneous data.** Let $m = 20$ and $|\mathfrak{D}_j| = 500$ for the synthetic dataset. We explore the impact of data heterogeneity by generating inputs with covariate shifts and explore the impact of model heterogeneity by generating outputs with response shifts. The right of Figure 3 illustrates: 1) Compared to DKRR, `FedNewton` remarkably reduce MSE when the heterogeneity is small. But it enlarges the errors from heterogeneous data when the heterogeneity is bigger than a threshold, i.e., $\Delta_{\mathfrak{D}_j} \approx 0.466$. 2) For the benign data heterogeneous settings, more communications for `FedNewton` lead to better approximation to the exact KRR, while the gap between DKRR and KRR still exists. 3) When data heterogeneity is large, `FedNewton` is more sensitive to data heterogeneity than DKRR, and more communications hurt the predictive accuracy. In the line of federated learning, the data heterogeneity is common due to different data distributions while the model heterogeneity is usually small. The left of Figure 4 shows that 1) Model heterogeneity decreases the predictive performance for all methods. 2) More communications lead to better approximation to

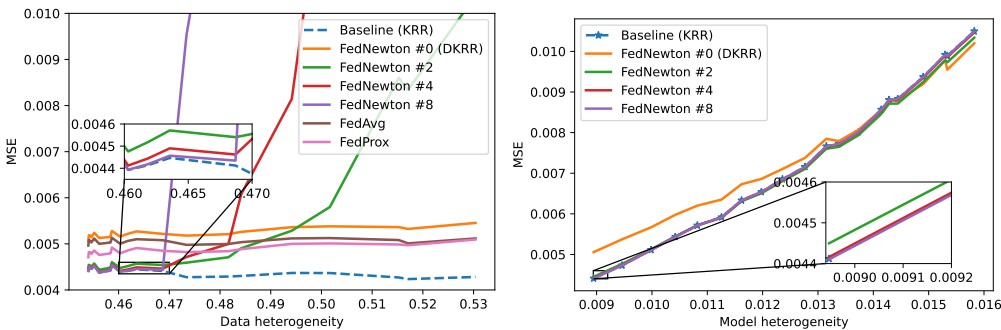

Figure 4: Impact of data heterogeneity (left) and model heterogeneity (right) on the synthetic dataset. We empirically estimate data heterogeneity by $\Delta_{\mathfrak{D}_j} = [\boldsymbol{\Phi}_{\mathcal{D}}^\top \boldsymbol{\Phi}_{\mathcal{D}} - \boldsymbol{\Phi}_{\mathfrak{D}_j}^\top \boldsymbol{\Phi}_{\mathfrak{D}_j}]$, and model heterogeneity by $\Delta_{f_j} = \frac{1}{|\mathfrak{D}_j|} \sum_{i=1}^{|\mathfrak{D}_j|} [f^*(\boldsymbol{x}_i) - f_j^*(\boldsymbol{x}_i)]$.

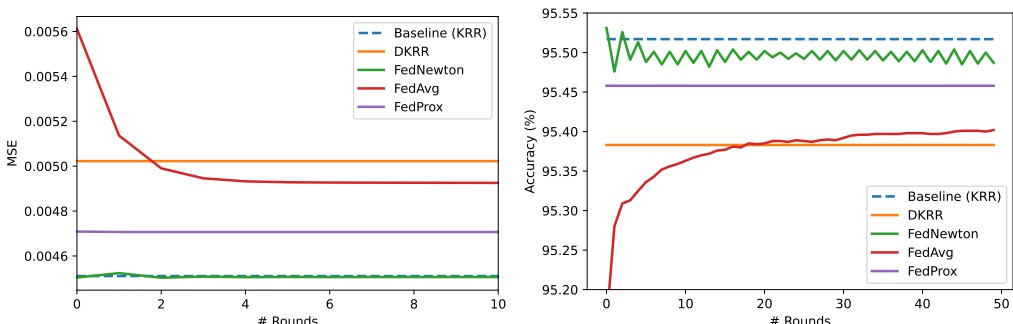

Figure 5: Predictive performance of `FedNewton`, FedAvg and FedProx on heterogeneous synthetic dataset (left) and MNIST (right).

KRR when model heterogeneity is small. 3) The performance of all methods is similarly poor when model heterogeneity is bigger than $0.0135$ and all models finally get similar bad results when model heterogeneity is large enough. These observations coincide with Theorem 4.

**Iterations of `FedNewton` and first-order methods.** We use heterogeneous dataset for iterations, i.e. the synthetic dataset with $\alpha = 0.01$ and $\gamma = 0.001$ and the MNIST dataset partitioned by a Dirichlet distribution $\text{Dir}_K(0.5)$. The last two in Figure 4 reports the generalization performance on heterogeneous data in terms of the communication rounds. We find that: 1) With a few iterations, `FedNewton` converges to KRR on the entire data, outperforming the divide-and-conquer and first-order methods. 2) Since local models are initialized by the closed-form solutions, FedProx converges very fast $t = 1$ and then updates slowly. The performance of FedProx is better than DKRR and FedAvg but worse than `FedNewton`. 3) Compared to FedProx and `FedNewton`, the convergence of FedAvg is slow and achieves the performance between DKRR and FedProx.

## 7.2 EVALUATION RESULTS ON REAL DATASETS

We compared related federated learning algorithms on both original datasets and non-iid datasets partitioned by a Dirichlet distribution $\text{Dir}_K(0.5)$. After partitioning with a Dirichlet distribution, the labels and the number of local samples on datasets are very unbalanced that decrease the generalization ability of federated learning algorithms. We report the classification accuracy in Table 3 for several public classification datasets, illustrating that:

1) The proposed `FedNewton` remarkably outperforms the compared methods on the original datasets, and more iterations improves the generalization performance. This observation coincides with results in Theorem 3.

Table 2: Data statistics and hyperparameter settings.

| Dataset | Task | $|\mathcal{D}|$ | $d$ | classes | kernel parameter $\sigma^2$ | $\lambda$ | $\lambda_{\text{prox}}$ | $\text{Dir}_K(\alpha)$ | learning rate $\gamma$ |
|---|---|---|---|---|---|---|---|---|---|
| synthetic | regression | 10000 | 10 | 1 | 0.1 | 1e-06 | 7e-07 | 1 | 0.001 |
| dna | multiclass | 2000 | 180 | 3 | 0.001 | 1e-07 | 1e-08 | 1 | 0.001 |
| letter | multiclass | 15000 | 16 | 26 | 1 | 0.001 | 0.001 | 0.5 | 0.001 |
| pendigits | multiclass | 7494 | 16 | 10 | 0.01 | 0.0001 | 0.0001 | 1 | 0.0001 |
| satimage | multiclass | 4435 | 36 | 6 | 1 | 0.001 | 0.001 | 1 | 0.001 |
| Sensorless | multiclass | 58509 | 48 | 11 | 10 | 1e-06 | 1e-07 | 1 | 0.001 |
| shuttle | multiclass | 43500 | 48 | 11 | 10 | 0.001 | 0.001 | 0.5 | 0.001 |
| usps | multiclass | 7291 | 256 | 10 | 0.1 | 0.0001 | 0.0001 | 0.5 | 0.001 |
| mnist | multiclass | 60000 | 784 | 10 | 0.1 | 1e-05 | 7e-07 | 0.5 | 0.001 |

Table 3: Classification accuracy (%) for classification datasets. We bold the results with the best method and underline the ones that are not significantly worse than the best one.

| Dataset | Compared methods | | | FedNewton | | | |
|---|---|---|---|---|---|---|---|
| | DKRR | FedAvg | FedProx | # 1 | # 2 | # 4 | # 8 |
| dna | 90.91±0.50 | 91.09±0.42 | 89.42±6.98 | **92.23±0.53** | 91.96±0.48 | 92.02±0.40 | 88.19±11.58 |
| letter | 77.18±0.12 | 77.11±0.17 | 77.17±0.12 | 77.30±0.12 | 77.30±0.12 | **77.30±0.12** | 77.30±0.12 |
| pendigits | 97.12±0.09 | 97.12±0.11 | 97.12±0.10 | 97.29±0.13 | **97.31±0.10** | 97.31±0.11 | 97.23±0.31 |
| satimage | 87.70±0.17 | 87.84±0.08 | 87.74±0.11 | **88.49±0.19** | 88.26±0.17 | 88.31±0.14 | 88.31±0.15 |
| Sensorless | 96.81±0.12 | 96.87±0.14 | 96.84±0.13 | **97.32±0.11** | 96.87±0.14 | 96.44±0.17 | 84.43±1.20 |
| shuttle | 98.46±0.06 | 98.53±0.08 | 98.51±0.07 | **98.54±0.07** | 98.51±0.07 | 98.50±0.07 | 98.44±0.16 |
| usps | 92.95±0.10 | 92.95±0.12 | 92.95±0.12 | **93.49±0.18** | 93.24±0.13 | 93.28±0.14 | 93.30±0.15 |
| mnist | 95.38±0.12 | 95.40±0.13 | 95.46±0.11 | **95.53±0.13** | 95.48±0.13 | 95.49±0.13 | 95.48±0.12 |

2) The predictive accuracies of all federated learning methods in the heterogeneous setting are worse than ones in the original case, but `FedNewton` approaches still achieve the optimal results on the most datasets.

3) Similar to Figure 4, `FedNewton` with more iterations are more sensitive to the heterogeneity and more iterations hurts the generalization performance. The reason is the number of iterations augments the federated error when $\Upsilon > 1$ due to large data heterogeneity.

# PROOFS

## 7.3 PRELIMINARIES

Since KRR has closed-form solutions, the intermediate estimators $\bar{f}^t_{\mathcal{D},\lambda}, f_{\mathcal{D},\lambda}, f_\lambda, f^*$ in error decomposition can be represented by the redirection operators and their adjoint operators. In this section, we first provide useful linear operators associated with kernel $K$. Then, we measure the similarities between empirical and expected covariance operators via concentration inequalities.

**Definition 2** (Operators with kernel $K$ in terms of the global distribution $\rho_{X \times Y}$)**.** *For any $\boldsymbol{x} \in \mathcal{X}, g \in L^2(\mathbb{P}), \phi : \mathcal{X} \to \mathcal{H}_K$ and $\boldsymbol{\beta} \in \mathcal{H}_K$, we define the following expected operators*

- $S : \mathcal{H}_K \to L^2(\mathbb{P}), \quad (S\boldsymbol{\beta})(\boldsymbol{x}) = \langle \boldsymbol{\beta}, \phi(\boldsymbol{x}) \rangle.$

- $S^* : L^2(\mathbb{P}) \to \mathcal{H}_K, \quad S^* g = \int_X \phi(\boldsymbol{x}) g(\boldsymbol{x}) d\rho_X(\boldsymbol{x}).$

- $L : L^2(\mathbb{P}) \to L^2(\mathbb{P}), \quad L = SS^*, \quad \text{such that } (Lg)(\cdot) = \int_X \langle \phi(\cdot), \phi(\boldsymbol{x}) \rangle g(\boldsymbol{x}) d\rho_X(\boldsymbol{x}).$

- $C : \mathcal{H}_K \to \mathcal{H}_K, \quad C = S^* S, \quad \text{such that } C\boldsymbol{\beta} = \int_X \langle \boldsymbol{\beta}, \phi(\boldsymbol{x}) \rangle \phi(\boldsymbol{x}) d\rho_X(\boldsymbol{x}).$

**Definition 3** (Empirical operators on the global dataset $\mathcal{D}$ and local datasets $\mathfrak{D}_j$)**.** *For any $\phi : \mathcal{X} \to \mathcal{H}_K$ and $\boldsymbol{\beta} \in \mathcal{H}_K$, we define the following empirical operators*

- $S_{\mathcal{D}} : \mathcal{H}_K \to \mathbb{R}^{|\mathcal{D}|}, \quad S_{\mathcal{D}}\boldsymbol{\beta} = \left( \langle \boldsymbol{\beta}, \phi(\boldsymbol{x}_i) \rangle \right)_{i=1}^{|\mathcal{D}|} \in \mathbb{R}^{|\mathcal{D}|}, \quad \forall (\boldsymbol{x}_i, y_i) \in \mathcal{D}.$

- $S_{\mathcal{D}}^* : \mathbb{R}^{|\mathcal{D}|} \to \mathcal{H}_K$,   $S_{\mathcal{D}}^* \alpha = \frac{1}{|\mathcal{D}|} \sum_{i=1}^{|\mathcal{D}|} \phi(\boldsymbol{x}_i) \alpha_i \in \mathcal{H}_K$,   $\forall (\boldsymbol{x}_i, y_i) \in \mathcal{D}, \; \alpha \in \mathbb{R}^{|\mathcal{D}|}$.

- $C_{\mathcal{D}} : \mathcal{H}_K \to \mathcal{H}_K$,   $C_{\mathcal{D}} = S_{\mathcal{D}}^* S_{\mathcal{D}}$, such that  $C_{\mathcal{D}} \, \boldsymbol{\beta} = \frac{1}{|\mathcal{D}|} \sum_{i=1}^{|\mathcal{D}|} \langle \boldsymbol{\beta}, \phi(\boldsymbol{x}_i) \rangle \phi(\boldsymbol{x}_i)$,   $\forall (\boldsymbol{x}_i, y_i) \in \mathcal{D}$.

- $S_{\mathfrak{D}_j} : \mathcal{H}_K \to \mathbb{R}^{|\mathfrak{D}_j|}$,   $S_{\mathfrak{D}_j} \boldsymbol{\beta} = \left( \langle \boldsymbol{\beta}, \phi(\boldsymbol{x}_i) \rangle \right)_{i=1}^{|\mathfrak{D}_j|} \in \mathbb{R}^{|\mathfrak{D}_j|}$,   $\forall (\boldsymbol{x}_i, y_i) \in \mathfrak{D}_j$.

- $S_{\mathfrak{D}_j}^* : \mathbb{R}^{|\mathfrak{D}_j|} \to \mathcal{H}_K$,   $S_{\mathfrak{D}_j}^* \alpha = \frac{1}{|\mathfrak{D}_j|} \sum_{i=1}^{|\mathfrak{D}_j|} \phi(\boldsymbol{x}_i) \alpha_i \in \mathcal{H}_K$,   $\forall (\boldsymbol{x}_i, y_i) \in \mathfrak{D}_j$.

- $C_{\mathfrak{D}_j} : \mathcal{H}_K \to \mathcal{H}_K$,   $C_{\mathfrak{D}_j} = S_{\mathfrak{D}_j}^* S_{\mathfrak{D}_j}$,  such that  $C_{\mathfrak{D}_j} \, \boldsymbol{\beta} = \frac{1}{|\mathfrak{D}_j|} \sum_{i=1}^{|\mathfrak{D}_j|} \langle \boldsymbol{\beta}, \phi(\boldsymbol{x}_i) \rangle \phi(\boldsymbol{x}_i)$,   $\forall (\boldsymbol{x}_i, y_i) \in \mathfrak{D}_j$.

Here, we denote $S$ the inclusion operator and $S_{\mathcal{D}}, S_{\mathfrak{D}_j}$ the sampling operator, while $S^*, S_{\mathcal{D}}^*, S_{\mathfrak{D}_j}^*$ are their adjoint operators. Note that $C : \mathcal{H}_K \to \mathcal{H}_K$ is the covariance operator given by $S^* S$, and the integral operator $L : L^2(\mathbb{P}) \to L^2(\mathbb{P})$ given by $SS^*$. The kernel matrix $\mathbf{K}_{\mathcal{D}}, \mathbf{K}_{\mathfrak{D}}$ and the covariance matrix $C_{\mathcal{D}}, C_{\mathfrak{D}_j}$ are the empirical counterparts of the integral operator $L$ and the covariance operator $C$, respectively. Using Singular Value Decomposition shows that $L$ and $C$ have the same eigenvalues, and the corresponding eigenvectors are closely related Rosasco et al. (2010). Those kernels-related operators are widely used in the proof of optimal learning theory for standard KRR. Assuming the kernel is bounded $K(\boldsymbol{x}, \boldsymbol{x}') \leq \kappa^2$, the integral operator $L$ and the covariance operator $C$ are positive trace class operators (and hence compact) and bounded by $\|L\| = \|C\| \leq \kappa^2$. For any function $f \in \mathcal{H}_K$, the estimator $f \in L^2(\mathbb{P})$ is obtained by kernel trick. Thus, for $f(\boldsymbol{x}) = \langle \boldsymbol{w}, \phi(\boldsymbol{x}) \rangle$, the RKHS norm can be related to the $L^2(\mathbb{P})$-norm by $C^{1/2}$ Bauer et al. (2007):

$$\|f\|_2 = \|Sf\|_2 = \|C^{1/2} \boldsymbol{w}\|_K, \quad \forall \boldsymbol{w} \in \mathcal{H}_K, \; f \in L^2(\mathbb{P}). \tag{8}$$

**Remark 8.** *With the assumption $K(\boldsymbol{x}, \boldsymbol{x}') \leq \kappa^2$, the integral operator $L$ is trace class Caponnetto & De Vito (2007) and $C, C_{\mathcal{D}}, C_{\mathfrak{D}_j}$ are finite dimensional. Moreover we have that $L = SS^*, C = S^* S, C_{\mathcal{D}} = S_{\mathcal{D}}^* S_{\mathcal{D}}$ and $C_{\mathfrak{D}_j} = S_{\mathfrak{D}_j}^* S_{\mathfrak{D}_j}$. Finally $L, C, C_{\mathcal{D}}, C_{\mathfrak{D}_j}$ are self-adjoint and positive operators, with spectrum is $[0, \kappa^2]$.*

**Proposition 2** (Cordes Inequality Fujii et al. (1993))**.** *Let $A, B$ two positive semi-definite bounded linear operators on a separable Hilbert space. Then*

$$\|A^s B^s\| \leq \|AB\|^s, \qquad when \quad 0 \leq s \leq 1.$$

Here, we use Proposition 2 to obtain the inequality $\|(A + \lambda I)^{-1/2} (B + \lambda)^{1/2}\| \leq \|(A + \lambda I)^{-1}(B + \lambda)\|^{1/2}$ for linear operators $C, C_j, C_{\mathcal{D}}, C_{\mathfrak{D}_j}$, and $L$.

**Proposition 3** (Lemma 2 in Smale & Zhou (2007))**.** *Let $\mathcal{L}$ be a separable Hilbert space and $\{\xi_1, \cdots, \xi_n\}$ be a sequence of i.i.d random variables in $\mathcal{L}$. Assume the bound be $\|\xi_i\| \leq \widetilde{M} \leq \infty$ and the variance be $\tilde{\sigma}^2 = \mathbb{E}(\|\xi_i - \mathbb{E}(\xi_i)\|^2)$ for any $i \in [n]$. For any $\delta \in (0, 1)$, with confidence $1 - \delta$,*

$$\left\| \frac{1}{n} \sum_{i=1}^{n} \xi_i - \mathbb{E}(\xi_i) \right\| \leq \frac{2\widetilde{M} \log(2/\delta)}{n} + \sqrt{\frac{2\tilde{\sigma}^2 \log(2/\delta)}{n}}. \tag{9}$$

The above Bernstein's inequality is the key to analyzing the relationship between the empirical random vector and its expected counterpart, which is used to prove Lemma 1. The above Bernstein's inequality for random vectors was provided in Smale & Zhou (2007); Rudi & Rosasco (2017) and later was extended to the random operator case in Theorem 7.3.1 in Tropp (2012) and Lemma 24 in Lin & Cevher (2020).

**Lemma 1.** *Given $K(\boldsymbol{x}, \boldsymbol{x}') = \langle \phi(\boldsymbol{x}), \phi(\boldsymbol{x}') \rangle_K$, let $\phi(\cdot)$ be i.i.d random vectors on a separable Hilbert space $\mathcal{H}_K$ such that $C, C_{\mathcal{D}}, C_{\mathfrak{D}_j}$ are trace class. Then for any $\delta \in (0, 1)$ with the probability*

*at least $1 - \delta$, the following holds*

$$\left\|(C + \lambda I)^{-1/2}(C - C_{\mathcal{D}})(C + \lambda I)^{-1/2}\right\| \leq \mathcal{R}_{\mathcal{D},\lambda} \leq \frac{2\kappa^2 \log(2/\delta)}{\lambda|\mathcal{D}|} + \sqrt{\frac{2(\kappa^2 + 1)\log(2/\delta)}{\lambda|\mathcal{D}|}},$$

$$\left\|(C_j + \lambda I)^{-1/2}(C_j - C_{\mathfrak{D}_j})(C_j + \lambda I)^{-1/2}\right\| \leq \mathcal{R}_{\mathfrak{D}_j,\lambda} \leq \frac{2\kappa^2 \log(2/\delta)}{\lambda|\mathfrak{D}_j|} + \sqrt{\frac{2(\kappa^2 + 1)\log(2/\delta)}{\lambda|\mathfrak{D}_j|}},$$

$$(10)$$

*where $\mathcal{R}_{\mathcal{D},\lambda} = \left\|(C + \lambda I)^{-1}(C - C_{\mathcal{D}})\right\|$ and $\mathcal{R}_{\mathfrak{D}_j,\lambda} = \left\|(C_j + \lambda I)^{-1}(C_j - C_{\mathfrak{D}_j})\right\|$.*

*Proof.* We first prove the lower bound for $\mathcal{R}_{\mathcal{D},\lambda}$. Using the Cauchy-Schwarz inequality, we have

$$
\begin{aligned}
&\left\|(C + \lambda I)^{-1/2}(C - C_{\mathcal{D}})(C + \lambda I)^{-1/2}\right\| \\
&= \left\|(C + \lambda I)^{-1/2}(C - C_{\mathcal{D}})^{1/2}(C - C_{\mathcal{D}})^{1/2}(C + \lambda I)^{-1/2}\right\| \\
&\leq \left\|(C + \lambda I)^{-1/2}(C - C_{\mathcal{D}})^{1/2}\right\|^2.
\end{aligned}
\tag{11}
$$

Recall that the norm on a matrix or operator $A$ can be defined By

$$\|A\| := \sup_x \frac{\|Ax\|_2}{\|x\|_2}.$$

For $K > 1$ and a nonzero vector $x$, we get

$$\|A^k x\|_2 = \|AA^{k-1}x\|_2 \leq \|A\|\|A^{k-1}x\|_2 \leq \cdots \leq \|A\|^k \|x\|_2.$$

Therefore, it holds $\frac{\|A^k x\|_2}{\|x\|_2} \leq \|A\|^k$ and thus

$$\|A^k\| = \sup_x \frac{\|A^k x\|_2}{\|x\|_2} \leq \|A\|^k. \tag{12}$$

Assuming $A = (C + \lambda I)^{-1/2}$ and substituting equation 12 to equation 11, we get

$$\left\|(C + \lambda I)^{-1/2}(C - C_{\mathcal{D}})(C + \lambda I)^{-1/2}\right\| \leq \left\|(C + \lambda I)^{-1}(C - C_{\mathcal{D}})\right\| = \mathcal{R}_{\mathcal{D},\lambda}.$$

Then, we prove the upper bound for $\mathcal{R}_{\mathcal{D},\lambda}$. Let $\xi = (C + \lambda I)^{-1}\phi(\boldsymbol{x}) \otimes \phi(\boldsymbol{x})$, thus we have

$$\mathbb{E}(\xi) = (C + \lambda I)^{-1}\mathbb{E}[\phi(\boldsymbol{x}) \otimes \phi(\boldsymbol{x})] = (C + \lambda I)^{-1}C,$$

$$\frac{1}{|\mathcal{D}|}\sum_{i=1}^{|\mathcal{D}|}\xi_i = \frac{1}{|\mathcal{D}|}\sum_{i=1}^{|\mathcal{D}|}(C + \lambda I)^{-1}[\phi(\boldsymbol{x}_i) \otimes \phi(\boldsymbol{x}_i)] = (C + \lambda I)^{-1}C_{\mathcal{D}}.$$

The left of the desired inequality becomes

$$\left\|(C + \lambda I)^{-1}(C - C_{\mathcal{D}})\right\| = \left\|\mathbb{E}(\xi) - \frac{1}{|\mathcal{D}|}\sum_{i=1}^{|\mathcal{D}|}\xi_i\right\|.$$

Note that

$$\|(C + \lambda I)^{-1/2}\phi(\boldsymbol{x})\|^2 \leq \kappa^2 \lambda^{-1}.$$

To use Bernstein's inequality (Proposition 3), we need to bound $\|\xi\|$ and $\mathbb{E}\|\xi\|^2$ as follows

$$\|\xi\| = \|\langle (C + \lambda I)^{-1}\phi(\boldsymbol{x}), \phi(\boldsymbol{x})\rangle\| \leq \|(C + \lambda I)^{-1/2}\phi(\boldsymbol{x})\|^2 \leq \kappa^2 \lambda^{-1}.$$

$$
\begin{aligned}
\mathbb{E}\|\xi - \mathbb{E}(\xi)\|^2 &= \left\|\mathbb{E}\left[\langle (C + \lambda I)^{-1}\phi(\boldsymbol{x}), \phi(\boldsymbol{x})\rangle(C + \lambda I)^{-1}\phi(\boldsymbol{x}) \otimes \phi(\boldsymbol{x})\right] - C_\lambda^{-2}C^2\right\| \\
&\leq \kappa^2 \lambda^{-1}\left\|\mathbb{E}\left[(C + \lambda I)^{-1}\phi(\boldsymbol{x}) \otimes \phi(\boldsymbol{x})\right]\right\| + \|C_\lambda^{-2}C^2\| \\
&\leq \kappa^2 \lambda^{-1}\|C_\lambda^{-1}C\| + 1 \leq \kappa^2 \lambda^{-1} + 1 \leq (\kappa^2 + 1)\lambda^{-1}.
\end{aligned}
$$

Substituting the above identities to Bernstein's inequality equation 9, we obtain the upper bound for $\mathcal{R}_{\mathcal{D},\lambda}$.

The lower and upper bounds can be proven with similar proof techniques. $\qquad\square$

**Lemma 2** (Proposition 8 Rudi & Rosasco (2017))**.** *Let $\lambda > 0$. We define the following quantities*

$$\mathcal{P}_{\mathcal{D},\lambda} := \left\| (C_{\mathcal{D}} + \lambda I)^{-1}(C + \lambda I) \right\|, \quad \mathcal{P}_{\mathfrak{D}_j,\lambda} := \left\| (C_{\mathfrak{D}_j} + \lambda I)^{-1}(C_j + \lambda I) \right\|.$$

*Then, there exists the following properties*

$$\mathcal{P}_{\mathcal{D},\lambda} \leq \frac{1}{1-\beta}, \quad \mathcal{P}_{\mathfrak{D}_j,\lambda} \leq \frac{1}{1-\beta},$$

*with*

$$\beta = \lambda_{max} \left[ (C + \lambda I)^{-1/2}(C - C_{\mathcal{D}})(C + \lambda I)^{-1/2} \right].$$

*Note that, $\beta \leq \frac{\lambda_{max}(C)}{\lambda_{max}+\lambda} < 1$.*

### 7.4 Error Decomposition for FedNewton

For Newton-based federated learning, there holds the following error decompositions

$$\|\bar{f}^t_{\mathcal{D},\lambda} - f^*\| \leq \|\bar{f}^t_{\mathcal{D},\lambda} - f_{\mathcal{D},\lambda}\| + \|f_{\mathcal{D},\lambda} - f^*\|. \tag{13}$$

Here, the federated error term $\|\bar{f}^t_{\mathcal{D},\lambda} - f_{\mathcal{D},\lambda}\|$ is also the key to analyzing the generalization of second-order optimization based federated learning FedNewton.

*Proof of Theorem 1.* For any function $f(\boldsymbol{x}) = \langle \boldsymbol{w}, \phi(\boldsymbol{x}) \rangle_K$, the $\mathcal{H}_K$-norm can be related to the $L^2(\mathbb{P})$-norm by the inclusion $S$ Bauer et al. (2007)

$$\|f\|_2 = \|S\boldsymbol{w}\|_K = \|S(C + \lambda I)^{-1/2}(C + \lambda I)^{1/2}\boldsymbol{w}\|_K \leq \|(C + \lambda I)^{1/2}\boldsymbol{w}\|_K.$$

Therefore, one can prove

$$\|\bar{f}^t_{\mathcal{D},\lambda} - f_{\mathcal{D},\lambda}\|_2 \leq \|(C + \lambda I)^{1/2}(\bar{\boldsymbol{w}}^t_{\mathcal{D},\lambda} - \boldsymbol{w}_{\mathcal{D},\lambda})\|_K. \tag{14}$$

From equation 5, we have

$$\bar{\boldsymbol{w}}^t_{\mathcal{D},\lambda} = \bar{\boldsymbol{w}}^{t-1}_{\mathcal{D},\lambda} - \sum_{j=1}^{m} p_j \boldsymbol{H}^{-1}_{\mathfrak{D}_j,\lambda} \boldsymbol{g}^{t-1}_{\mathcal{D},\lambda}$$

$$= \bar{\boldsymbol{w}}^{t-1}_{\mathcal{D},\lambda} - \sum_{j=1}^{m} p_j (C_{\mathfrak{D}_j} + \lambda I)^{-1} \left[ (C_{\mathcal{D}} + \lambda I)\bar{\boldsymbol{w}}^{t-1}_{\mathcal{D},\lambda} - S^*_{\mathcal{D}}\boldsymbol{y}_{\mathcal{D}} \right]$$

$$= \sum_{j=1}^{m} p_j (C_{\mathfrak{D}_j} + \lambda I)^{-1}(C_{\mathfrak{D}_j} - C_{\mathcal{D}})\bar{\boldsymbol{w}}^{t-1}_{\mathcal{D},\lambda} + \sum_{j=1}^{m} p_j (C_{\mathfrak{D}_j} + \lambda I)^{-1}S^*_{\mathcal{D}}\boldsymbol{y}_{\mathcal{D}}$$

$$= \sum_{j=1}^{m} p_j (C_{\mathfrak{D}_j} + \lambda I)^{-1}(C_{\mathfrak{D}_j} - C_{\mathcal{D}})\bar{\boldsymbol{w}}^{t-1}_{\mathcal{D},\lambda} + \sum_{j=1}^{m} p_j (C_{\mathfrak{D}_j} + \lambda I)^{-1}(C_{\mathcal{D}} + \lambda I)\boldsymbol{w}_{\mathcal{D},\lambda}.$$

And then, one can obtain

$$\bar{\boldsymbol{w}}^t_{\mathcal{D},\lambda} - \boldsymbol{w}_{\mathcal{D},\lambda}$$

$$= \sum_{j=1}^{m} p_j (C_{\mathfrak{D}_j} + \lambda I)^{-1}(C_{\mathfrak{D}_j} - C_{\mathcal{D}})\bar{\boldsymbol{w}}^{t-1}_{\mathcal{D},\lambda} + \sum_{j=1}^{m} p_j (C_{\mathfrak{D}_j} + \lambda I)^{-1}(C_{\mathcal{D}} + \lambda I)\boldsymbol{w}_{\mathcal{D},\lambda} - \boldsymbol{w}_{\mathcal{D},\lambda}$$

$$= \sum_{j=1}^{m} p_j (C_{\mathfrak{D}_j} + \lambda I)^{-1}(C_{\mathfrak{D}_j} - C_{\mathcal{D}})\bar{\boldsymbol{w}}^{t-1}_{\mathcal{D},\lambda} + \sum_{j=1}^{m} p_j (C_{\mathfrak{D}_j} + \lambda I)^{-1}(C_{\mathcal{D}} - C_{\mathfrak{D}_j})\boldsymbol{w}_{\mathcal{D},\lambda}$$

$$= \sum_{j=1}^{m} p_j (C_{\mathfrak{D}_j} + \lambda I)^{-1}(C_{\mathfrak{D}_j} - C_{\mathcal{D}})(\bar{\boldsymbol{w}}^{t-1}_{\mathcal{D},\lambda} - \boldsymbol{w}_{\mathcal{D},\lambda}).$$

We then estimate the federated error by

$$(C + \lambda I)^{1/2}(\bar{\boldsymbol{w}}_{\mathcal{D},\lambda}^t - \boldsymbol{w}_{\mathcal{D},\lambda})$$

$$= \sum_{j=1}^m p_j (C + \lambda I)^{1/2}(C_{\mathfrak{D}_j} + \lambda I)^{-1}(C_{\mathfrak{D}_j} - C_{\mathcal{D}})(\bar{\boldsymbol{w}}_{\mathcal{D},\lambda}^{t-1} - \boldsymbol{w}_{\mathcal{D},\lambda})$$

$$= \sum_{j=1}^m p_j (C + \lambda I)^{1/2}(C_{\mathfrak{D}_j} + \lambda I)^{-1}(C_{\mathfrak{D}_j} - C_j + C_j - C + C - C_{\mathcal{D}})(\bar{\boldsymbol{w}}_{\mathcal{D},\lambda}^{t-1} - \boldsymbol{w}_{\mathcal{D},\lambda})$$

$$= \sum_{j=1}^m p_j (C + \lambda I)^{1/2}(C_j + \lambda I)^{-1/2}(C_j + \lambda I)^{1/2}(C_{\mathfrak{D}_j} + \lambda I)^{-1}(C_j + \lambda I)^{1/2}$$

$$\quad (C_j + \lambda I)^{-1/2}(C_{\mathfrak{D}_j} - C_j)(C_j + \lambda I)^{-1/2}(C_j + \lambda I)^{1/2}(C + \lambda I)^{-1/2}(C + \lambda I)^{1/2}(\bar{\boldsymbol{w}}_{\mathcal{D},\lambda}^{t-1} - \boldsymbol{w}_{\mathcal{D},\lambda})$$

$$\quad + \sum_{j=1}^m p_j (C + \lambda I)^{1/2}(C_j + \lambda I)^{-1/2}(C_j + \lambda I)^{1/2}(C_{\mathfrak{D}_j} + \lambda I)^{-1}(C_j + \lambda I)^{1/2}$$

$$\quad (C_j + \lambda I)^{-1/2}(C_j - C)(C_j + \lambda I)^{-1/2}(C_j + \lambda I)^{1/2}(C + \lambda I)^{-1/2}(C + \lambda I)^{1/2}(\bar{\boldsymbol{w}}_{\mathcal{D},\lambda}^{t-1} - \boldsymbol{w}_{\mathcal{D},\lambda})$$

$$\quad + \sum_{j=1}^m p_j (C + \lambda I)^{1/2}(C_j + \lambda I)^{-1/2}(C_j + \lambda I)^{1/2}(C_{\mathfrak{D}_j} + \lambda I)^{-1}(C_j + \lambda I)^{1/2}$$

$$\quad (C_j + \lambda I)^{-1/2}(C + \lambda I)^{1/2}(C + \lambda I)^{-1/2}(C - C_{\mathcal{D}})(C + \lambda I)^{-1/2}(C + \lambda I)^{1/2}(\bar{\boldsymbol{w}}_{\mathcal{D},\lambda}^{t-1} - \boldsymbol{w}_{\mathcal{D},\lambda}). \tag{15}$$

Note that, $\|(C + \lambda I)^{1/2}(C_j + \lambda I)^{-1/2}\| \leq \|I + (C_j + \lambda I)^{-1}(C - C_j)\|^{1/2} \leq \sqrt{1 + \frac{\Delta_{\mathfrak{D}_j}}{\lambda}}$, $\|(C_j + \lambda I)^{1/2}(C_{\mathfrak{D}_j} + \lambda I)^{-1}(C_j + \lambda I)^{1/2}\| \leq \mathcal{P}_{\mathfrak{D}_j,\lambda}$, $\|(C_j + \lambda I)^{1/2}(C + \lambda I)^{-1/2}\| \leq \|I + (C + \lambda I)^{-1}(C_j - C)\|^{1/2} \leq \sqrt{1 + \frac{\Delta_{\mathfrak{D}_j}}{\lambda}}$. Therefore, substituting these inequalities to equation 15 and from equation 14, there exists

$$\|\bar{f}_{\mathcal{D},\lambda}^t - f_{\mathcal{D},\lambda}\|_2$$

$$\leq \|(C + \lambda I)^{1/2}(\bar{\boldsymbol{w}}_{\mathcal{D},\lambda}^t - \boldsymbol{w}_{\mathcal{D},\lambda})\|_K$$

$$\leq \sum_{j=1}^m p_j \left(1 + \frac{\Delta_{\mathfrak{D}_j}}{\lambda}\right) \mathcal{P}_{\mathfrak{D}_j,\lambda} \mathcal{R}_{\mathfrak{D}_j,\lambda} \left\|(C + \lambda I)^{1/2}(\bar{\boldsymbol{w}}_{\mathcal{D},\lambda}^{t-1} - \boldsymbol{w}_{\mathcal{D},\lambda})\right\|_K$$

$$\quad + \sum_{j=1}^m p_j \left(1 + \frac{\Delta_{\mathfrak{D}_j}}{\lambda}\right) \mathcal{P}_{\mathfrak{D}_j,\lambda} \frac{\Delta_{\mathfrak{D}_j}}{\lambda} \left\|(C + \lambda I)^{1/2}(\bar{\boldsymbol{w}}_{\mathcal{D},\lambda}^{t-1} - \boldsymbol{w}_{\mathcal{D},\lambda})\right\|_K$$

$$\quad + \sum_{j=1}^m p_j \left(1 + \frac{\Delta_{\mathfrak{D}_j}}{\lambda}\right) \mathcal{P}_{\mathfrak{D}_j,\lambda} \mathcal{R}_{\mathcal{D},\lambda} \left\|(C + \lambda I)^{1/2}(\bar{\boldsymbol{w}}_{\mathcal{D},\lambda}^{t-1} - \boldsymbol{w}_{\mathcal{D},\lambda})\right\|_K \tag{16}$$

$$\leq \sum_{j=1}^m p_j \mathcal{P}_{\mathfrak{D}_j,\lambda} \left(1 + \frac{\Delta_{\mathfrak{D}_j}}{\lambda}\right) \left(\mathcal{R}_{\mathcal{D},\lambda} + \mathcal{R}_{\mathfrak{D}_j,\lambda} + \frac{\Delta_{\mathfrak{D}_j}}{\lambda}\right) \left\|(C + \lambda I)^{1/2}(\bar{\boldsymbol{w}}_{\mathcal{D},\lambda}^{t-1} - \boldsymbol{w}_{\mathcal{D},\lambda})\right\|_K$$

$$\leq \left(\sum_{j=1}^m p_j \mathcal{P}_{\mathfrak{D}_j,\lambda} \left(1 + \frac{\Delta_{\mathfrak{D}_j}}{\lambda}\right) \left(2\mathcal{R}_{\mathfrak{D}_j,\lambda} + \frac{\Delta_{\mathfrak{D}_j}}{\lambda}\right)\right)^t \left\|(C + \lambda I)^{1/2}(\bar{\boldsymbol{w}}_{\mathcal{D},\lambda}^0 - \boldsymbol{w}_{\mathcal{D},\lambda})\right\|_K.$$

Note that, $\mathcal{R}_{\mathcal{D},\lambda} \propto 1/|\mathcal{D}|$ and thus $\mathcal{R}_{\mathfrak{D}_j,\lambda} \leq \mathcal{R}_{\mathcal{D},\lambda}$. Combing the above inequality and equation 13, we prove the final result. $\qquad\square$

**Proposition 4.** *The following federated error bounds hold for oneshot federated learning:*

$$\|(C + \lambda I)^{1/2}(\bar{\boldsymbol{w}}_{\mathcal{D},\lambda}^0 - \boldsymbol{w}_{\mathcal{D},\lambda})\|_K$$

$$\leq \mathcal{P}_{\mathcal{D},\lambda} \sum_{j=1}^m p_j \left(2\mathcal{R}_{\mathfrak{D}_j,\lambda} + \frac{(1 + \mathcal{R}_{\mathfrak{D}_j,\lambda})\Delta_{\mathfrak{D}_j}}{\lambda}\right) \left\|(C + \lambda I)^{1/2}(\boldsymbol{w}_{\mathfrak{D}_j,\lambda} - \boldsymbol{w}_\lambda)\right\|_K, \tag{17}$$

*where $\Delta_{\mathfrak{D}_j} = \|C_j - C\|$.*

*Proof.* Note that, if $A, B$ are invertible operators on a Banach space, then there holds the equality

$$A^{-1} - B^{-1} = B^{-1}(B - A)A^{-1} = A^{-1}(B - A)B^{-1}.$$

From equation 2, using the facts $S_{\mathcal{D}}^* \boldsymbol{y}_{\mathcal{D}} = \sum_{j=1}^{m} p_j S_{\mathfrak{D}_j}^* \boldsymbol{y}_{\mathfrak{D}_j}$ and $A^{-1} - B^{-1} = A^{-1}(B - A)B^{-1}$, we have

$$\bar{\boldsymbol{w}}_{\mathcal{D},\lambda}^0 - \boldsymbol{w}_{\mathcal{D},\lambda}$$

$$= \sum_{j=1}^{m} p_j (\boldsymbol{\Phi}_{\mathfrak{D}_j}^\top \boldsymbol{\Phi}_{\mathfrak{D}_j} + \lambda I)^{-1} \boldsymbol{\Phi}_{\mathfrak{D}_j}^\top \boldsymbol{y}_{\mathfrak{D}_j} - (\boldsymbol{\Phi}_{\mathcal{D}}^\top \boldsymbol{\Phi}_{\mathcal{D}} + \lambda I)^{-1} \boldsymbol{\Phi}_{\mathcal{D}}^\top \boldsymbol{y}_{\mathcal{D}}$$

$$= \sum_{j=1}^{m} p_j (C_{\mathfrak{D}_j} + \lambda I)^{-1} S_{\mathfrak{D}_j}^* \boldsymbol{y}_{\mathfrak{D}_j} - (C_{\mathcal{D}} + \lambda I)^{-1} S_{\mathcal{D}}^* \boldsymbol{y}_{\mathcal{D}}$$

$$= \sum_{j=1}^{m} p_j [(C_{\mathfrak{D}_j} + \lambda I)^{-1} - (C_{\mathcal{D}} + \lambda I)^{-1}] S_{\mathfrak{D}_j}^* \boldsymbol{y}_{\mathfrak{D}_j}$$

$$= \sum_{j=1}^{m} p_j (C_{\mathcal{D}} + \lambda I)^{-1} (C_{\mathcal{D}} - C_{\mathfrak{D}_j}) \boldsymbol{w}_{\mathfrak{D}_j,\lambda}$$

$$= \sum_{j=1}^{m} p_j (C_{\mathcal{D}} + \lambda I)^{-1} (C_{\mathcal{D}} - C) \boldsymbol{w}_{\mathfrak{D}_j,\lambda} + \sum_{j=1}^{m} p_j (C_{\mathcal{D}} + \lambda I)^{-1} (C - C_{\mathfrak{D}_j}) \boldsymbol{w}_{\mathfrak{D}_j,\lambda}$$

$$= \sum_{j=1}^{m} p_j (C_{\mathcal{D}} + \lambda I)^{-1} (C_{\mathcal{D}} - C)(\boldsymbol{w}_{\mathfrak{D}_j,\lambda} - \boldsymbol{w}_\lambda) + \sum_{j=1}^{m} p_j (C_{\mathcal{D}} + \lambda I)^{-1} (C_{\mathcal{D}} - C) \boldsymbol{w}_\lambda$$

$$+ \sum_{j=1}^{m} p_j (C_{\mathcal{D}} + \lambda I)^{-1} (C - C_{\mathfrak{D}_j}) \boldsymbol{w}_{\mathfrak{D}_j,\lambda}$$

$$= \sum_{j=1}^{m} p_j (C_{\mathcal{D}} + \lambda I)^{-1} (C_{\mathcal{D}} - C)(\boldsymbol{w}_{\mathfrak{D}_j,\lambda} - \boldsymbol{w}_\lambda) + \sum_{j=1}^{m} p_j (C_{\mathcal{D}} + \lambda I)^{-1} (C - C_{\mathfrak{D}_j})(\boldsymbol{w}_{\mathfrak{D}_j,\lambda} - \boldsymbol{w}_\lambda).$$

$$(18)$$

The last step is due to the fact $\sum_{j=1}^{m} p_j C_{\mathcal{D}} = \sum_{j=1}^{m} p_j C_{\mathfrak{D}_j}$.

Combining equation 14 and equation 18, we have

$$\|\bar{f}_{\mathcal{D},\lambda}^0 - f_{\mathcal{D},\lambda}\|_2 \leq \|(C + \lambda I)^{1/2} (\bar{\boldsymbol{w}}_{\mathcal{D},\lambda}^0 - \boldsymbol{w}_{\mathcal{D},\lambda})\|_K$$

$$\leq \left\| \sum_{j=1}^{m} p_j (C + \lambda I)^{1/2} (C_{\mathcal{D}} + \lambda I)^{-1} (C_{\mathcal{D}} - C + C - C_{\mathfrak{D}_j})(\boldsymbol{w}_{\mathfrak{D}_j,\lambda} - \boldsymbol{w}_\lambda) \right\|.$$

$$(19)$$

Note that

$$(C + \lambda I)^{1/2} (C_{\mathcal{D}} + \lambda I)^{-1} (C_{\mathcal{D}} - C)$$

$$= (C + \lambda I)^{1/2} (C_{\mathcal{D}} + \lambda I)^{-1/2} (C_{\mathcal{D}} + \lambda I)^{-1/2} (C + \lambda I)^{1/2} (C + \lambda I)^{-1/2} (C_{\mathcal{D}} - C)(C + \lambda I)^{-1/2} (C + \lambda I)^{1/2}.$$

Using the inequality $\|(C + \lambda I)^{-1/2} (C_{\mathcal{D}} - C)(C + \lambda I)^{-1/2}\| \leq \mathcal{R}_{\mathcal{D},\lambda}$ from Lemma 1, we have

$$\|(C + \lambda I)^{1/2} (C_{\mathcal{D}} + \lambda I)^{-1} (C_{\mathcal{D}} - C)(\boldsymbol{w}_{\mathfrak{D}_j,\lambda} - \boldsymbol{w}_\lambda)\|$$

$$\leq \mathcal{P}_{\mathcal{D},\lambda} \|(C + \lambda I)^{-1/2} (C_{\mathcal{D}} - C)(C + \lambda I)^{-1/2} (C + \lambda I)^{1/2} (\boldsymbol{w}_{\mathfrak{D}_j,\lambda} - \boldsymbol{w}_\lambda)\| \qquad (20)$$

$$\leq \mathcal{P}_{\mathcal{D},\lambda} \mathcal{R}_{\mathcal{D},\lambda} \|(C + \lambda I)^{1/2} (\boldsymbol{w}_{\mathfrak{D}_j,\lambda} - \boldsymbol{w}_\lambda)\|.$$

Similarly, we have

$$(C + \lambda I)^{1/2}(C_{\mathcal{D}} + \lambda I)^{-1}(C - C_{\mathfrak{D}_j})$$

$$= (C + \lambda I)^{1/2}(C_{\mathcal{D}} + \lambda I)^{-1}(C - C_j + C_j - C_{\mathfrak{D}_j})$$

$$= (C + \lambda I)^{1/2}(C_{\mathcal{D}} + \lambda I)^{-1}(C + \lambda I)^{1/2}(C + \lambda I)^{-1/2}(C - C_j)(C + \lambda I)^{-1/2}(C + \lambda I)^{1/2}$$

$$+ (C + \lambda I)^{1/2}(C_{\mathcal{D}} + \lambda I)^{-1}(C + \lambda I)^{1/2}(C + \lambda I)^{-1/2}(C_j + \lambda I)^{1/2}$$

$$(C_j + \lambda I)^{-1/2}(C_j - C_{\mathfrak{D}_j})(C_j + \lambda I)^{-1/2}(C_j + \lambda I)^{1/2}(C + \lambda I)^{-1/2}(C + \lambda I)^{1/2}.$$

Using $\|(C + \lambda I)^{-1/2}(C_j + \lambda I)^{1/2}\| \leq \|I + (C + \lambda I)^{-1}(C_j - C)\|^{1/2} \leq 1 + \frac{\Delta_{\mathfrak{D}_j}}{\lambda}$, it holds

$$\|(C + \lambda I)^{1/2}(C_{\mathcal{D}} + \lambda I)^{-1}(C - C_{\mathfrak{D}_j})(\boldsymbol{w}_{\mathfrak{D}_j,\lambda} - \boldsymbol{w}_\lambda)\|$$

$$\leq \frac{\mathcal{P}_{\mathcal{D},\lambda}\Delta_{\mathfrak{D}_j}}{\lambda}\|(C + \lambda I)^{1/2}(\boldsymbol{w}_{\mathfrak{D}_j,\lambda} - \boldsymbol{w}_\lambda)\| + \mathcal{P}_{\mathcal{D},\lambda}\mathcal{R}_{\mathfrak{D}_j,\lambda}\left(1 + \frac{\Delta_{\mathfrak{D}_j}}{\lambda}\right)\|(C + \lambda I)^{1/2}(\boldsymbol{w}_{\mathfrak{D}_j,\lambda} - \boldsymbol{w}_\lambda)\|$$

$$\leq \mathcal{P}_{\mathcal{D},\lambda}\left(\mathcal{R}_{\mathfrak{D}_j,\lambda} + \frac{(1 + \mathcal{R}_{\mathfrak{D}_j,\lambda})\Delta_{\mathfrak{D}_j}}{\lambda}\right)\|(C + \lambda I)^{1/2}(\boldsymbol{w}_{\mathfrak{D}_j,\lambda} - \boldsymbol{w}_\lambda)\|.$$

$$(21)$$

Therefore, substituting equation 20 and equation 21 to equation 19, we have

$$\|\bar{f}_{\mathcal{D},\lambda}^0 - f_{\mathcal{D},\lambda}\| \leq \sum_{j=1}^m p_j \mathcal{P}_{\mathcal{D},\lambda}\left(\mathcal{R}_{\mathcal{D},\lambda} + \mathcal{R}_{\mathfrak{D}_j,\lambda} + \frac{(1 + \mathcal{R}_{\mathfrak{D}_j,\lambda})\Delta_{\mathfrak{D}_j}}{\lambda}\right)\|(C + \lambda I)^{1/2}(\boldsymbol{w}_{\mathfrak{D}_j,\lambda} - \boldsymbol{w}_\lambda)\|$$

$$\leq \mathcal{P}_{\mathcal{D},\lambda}\sum_{j=1}^m p_j\left(2\mathcal{R}_{\mathfrak{D}_j,\lambda} + \frac{(1 + \mathcal{R}_{\mathfrak{D}_j,\lambda})\Delta_{\mathfrak{D}_j}}{\lambda}\right)\|(C + \lambda I)^{1/2}(\boldsymbol{w}_{\mathfrak{D}_j,\lambda} - \boldsymbol{w}_\lambda)\|.$$

$$\square$$

## 7.5 ESTIMATING ERROR TERMS

### 7.5.1 ESTIMATING FEDERATED ERROR

From Lemma 1, Lemma 4, and equation 13, there are two error terms $\|\boldsymbol{w}_{\mathfrak{D}_j,\lambda} - \boldsymbol{w}_\lambda\|_K$ and $\|f_{\mathcal{D}_j,\lambda} - f_\lambda\|_2$ in federated error to be bounded. Using Bennett's inequality (Proposition 3), we first provide two useful lemmas.

**Lemma 3.** *Assume there exists $\kappa \geq 1$ such that $\|\phi(\boldsymbol{x})\|_K \leq \kappa$, $\forall \boldsymbol{x} \in \mathcal{X}$ and $|y| \leq B$. For $\delta \in (0, 1]$, the following holds with the probability at least $1 - \delta$*

$$\|(C + \lambda I)^{-1/2}(S_{\mathcal{D}}^* \boldsymbol{y}_{\mathcal{D}} - S^* f^*)\| \leq 2B\kappa \mathcal{A}_{\mathcal{D},\lambda} \log \frac{2}{\delta},$$

$$\|(C_j + \lambda I)^{-1/2}(S_{\mathfrak{D}_j}^* \boldsymbol{y}_{\mathfrak{D}_j} - S_j^* f_j^*)\| \leq 2B\kappa \mathcal{A}_{\mathfrak{D}_j,\lambda} \log \frac{2}{\delta}.$$

*where $C_j, S_j^*$ are operators defined on the local distribution $\rho_j$, and*

$$\mathcal{A}_{\mathcal{D},\lambda} := \frac{1}{|\mathcal{D}|\sqrt{\lambda}} + \sqrt{\frac{\mathcal{N}(\lambda)}{|\mathcal{D}|}}, \quad \mathcal{A}_{\mathfrak{D}_j,\lambda} := \frac{1}{|\mathfrak{D}_j|\sqrt{\lambda}} + \sqrt{\frac{\mathcal{N}(\lambda)}{|\mathfrak{D}_j|}}. \tag{22}$$

*Proof.* Let $\xi_i = (C + \lambda I)^{-1/2}\phi(\boldsymbol{x}_i)y_i$ in the Hilbert space $\mathcal{H}_K$. We see that

$$\frac{1}{|\mathcal{D}|}\sum_{i=1}^{|\mathcal{D}|}\xi_i = \frac{1}{n}\sum_{i=1}^n (C + \lambda I)^{-1/2}\phi(\boldsymbol{x}_i)y_i = (C + \lambda I)^{-1/2}S_{\mathcal{D}}\boldsymbol{y}_{\mathcal{D}},$$

$$\mathbb{E}\xi = \int_X (C + \lambda I)^{-1/2}\phi(\boldsymbol{x})f^*(\boldsymbol{x})d\rho_X(\boldsymbol{x}) = (C + \lambda I)^{-1/2}S^* f^*$$

Thus, the error term to bound can be stated as

$$\|(C + \lambda I)^{-1/2}(\widehat{S}_n^* \boldsymbol{y}_{\mathcal{D}} - S^* f^*)\| = \left\| \frac{1}{|\mathcal{D}|} \sum_{i=1}^{|\mathcal{D}|} \xi_i - \mathbb{E}\xi_i \right\|. \tag{23}$$

The rhs of the above identity can be bounded by Bennett's inequality (Proposition 3), thus we need to estimate $\|\xi_i - \mathbb{E}(\xi_i)\|$ and $\mathbb{E}\|\xi_i - \mathbb{E}(\xi_i)\|^2$ first.

We first recall the definition of effective dimension

$$\mathcal{N}(\lambda) = \mathbb{E} \langle \phi(\boldsymbol{x}), (C + \lambda I)^{-1}\phi(\boldsymbol{x}) \rangle_K = \int_X \|(C + \lambda I)^{-1}\phi(\boldsymbol{x})\|_K^2 \, d\rho_X(\boldsymbol{x}).$$

By Jensen's inequality, we thus have

$$\|\xi_i - \mathbb{E}(\xi_i)\| \leq \|(C + \lambda I)^{-1/2}\phi(\boldsymbol{x}_i)\||y_i| + \mathbb{E}\|(C + \lambda I)^{-1/2}\phi(\boldsymbol{x}_i)\||y_i| \leq 2B\kappa\lambda^{-1/2}. \tag{24}$$

Note that

$$\mathbb{E}\|\xi_i - \mathbb{E}(\xi_i)\|^2 \leq 2\int_X \|(C + \lambda I)^{-1/2}\phi(\boldsymbol{x}_i)\|^2|y_i|^2 d\rho_X(\boldsymbol{x})$$
$$\leq 2B^2 \int_X \|(C + \lambda I)^{-1/2}\phi(\boldsymbol{x}_i)\|^2 d\rho_X(\boldsymbol{x}) \leq 2B^2\mathcal{N}(\lambda). \tag{25}$$

Substituting equation 24 and equation 25 to equation 23, by Bennett's inequality (Proposition 3), we have

$$\|(C + \lambda I)^{-1/2}(S_{\mathcal{D}}^* \boldsymbol{y}_{\mathcal{D}} - S^* f^*)\| \leq \frac{2B\kappa \log(2/\delta)}{|\mathcal{D}|\sqrt{\lambda}} + 2\sqrt{\frac{B^2\mathcal{N}(\lambda)\log(2/\delta)}{|\mathcal{D}|}}.$$

Similarly, we derive the bound for $\|(C_j + \lambda I)^{-1/2}(S_{\mathfrak{D}_j}^* \boldsymbol{y}_{\mathfrak{D}_j} - S_{\mathfrak{D}_j}^* f_j^*)\|$. Thus, we prove the result.
$\square$

**Lemma 4** (From Theoreom 4 of Caponnetto & De Vito (2007)). *Assume there exists $\kappa \geq 1$ such that $\|\phi(\boldsymbol{x})\|_K \leq \kappa$, $\forall \boldsymbol{x} \in \mathcal{X}$. For $\delta \in (0, 1]$, the following holds with the probability at least $1 - \delta$*

$$\|(C + \lambda I)^{-1/2}(C - C_{\mathcal{D}})\| \leq 2\kappa(\kappa + 1)\mathcal{A}_{\mathcal{D},\lambda} \log \frac{2}{\delta},$$

$$\|(C_j + \lambda I)^{-1/2}(C_j - C_{\mathfrak{D}_j})\| \leq 2\kappa(\kappa + 1)\mathcal{A}_{\mathfrak{D}_j,\lambda} \log \frac{2}{\delta}.$$

The above lemma is a standard method for the difference between expected and empirical covariance operators $C - C_{\mathcal{D}}$ and $C_j - C_{\mathfrak{D}_j}$. Using a concentration inequality in Hilbert spaces, it have been proven in Caponnetto & De Vito (2007); Smale & Zhou (2007); Guo et al. (2017).

We define the expected estimators for local machines and centralized model as

$$\boldsymbol{w}_{j,\lambda} = \operatorname*{arg\,min}_{\boldsymbol{w} \in \mathcal{H}_K} \left\{ \int_X (\langle \boldsymbol{w}, \phi(\boldsymbol{x}) \rangle - f^*(\boldsymbol{x}))^2 d\rho_j(\boldsymbol{x}) + \lambda\|\boldsymbol{w}\|_K^2 \right\}$$

$$\boldsymbol{w}_{\lambda} = \operatorname*{arg\,min}_{\boldsymbol{w} \in \mathcal{H}_K} \left\{ \int_X (\langle \boldsymbol{w}, \phi(\boldsymbol{x}) \rangle - f^*(\boldsymbol{x}))^2 d\rho_X(\boldsymbol{x}) + \lambda\|\boldsymbol{w}\|_K^2 \right\}.$$

**Proposition 5.** *Assume $\|\phi(\boldsymbol{x})\|_K \leq \kappa$ and $|y| \leq B$. Under Assumption 2, for $\delta \in (0, 1/2)$, the following bound hold with the probability at least $1 - 2\delta$*

$$\|(C + \lambda I)^{1/2}(\boldsymbol{w}_{\mathfrak{D}_j,\lambda} - \boldsymbol{w}_{\lambda})\| \leq C_1\sqrt{1 + \frac{\Delta_{\mathfrak{D}_j}}{\lambda}} \mathcal{P}_{\mathfrak{D}_j,\lambda}\mathcal{A}_{\mathfrak{D}_j,\lambda} \log \frac{2}{\delta} + \frac{\kappa^2 R\Delta_{\mathfrak{D}_j}}{\lambda} + \Delta_{f_j}. \tag{26}$$

*where $C_1 = 2\kappa(B + 2\kappa^3 R)$.*

*Proof.* We introduce the intermediate estimators $\boldsymbol{w}_{j,\lambda} = (C_j + \lambda I)^{-1} S_j^* f_j^*$, where $S_j^*$ and $C_j$ are operators defined on the local distribution $\rho_j$. Then, it holds

$$\|(C + \lambda I)^{1/2}(\boldsymbol{w}_{\mathfrak{D}_j,\lambda} - \boldsymbol{w}_\lambda)\| \leq \|(C + \lambda I)^{1/2}(\boldsymbol{w}_{\mathfrak{D}_j,\lambda} - \boldsymbol{w}_{j,\lambda})\| + \|(C + \lambda I)^{1/2}(\boldsymbol{w}_{j,\lambda} - \boldsymbol{w}_\lambda)\| \tag{27}$$

where $\|\boldsymbol{w}_{\mathfrak{D}_j,\lambda} - \boldsymbol{w}_{j,\lambda}\|$ is the local variance and $\Delta_{f_j}$ is the model heterogeneity.

$$(C + \lambda I)^{1/2}(\boldsymbol{w}_{j,\lambda} - \boldsymbol{w}_\lambda)$$
$$=(C + \lambda I)^{1/2}\left[(C_j + \lambda I)^{-1} S_j^* f_j^* - (C + \lambda I)^{-1} S^* f^*\right]$$
$$=(C + \lambda I)^{1/2}\left[(C_j + \lambda I)^{-1} S_j^* f_j^* - (C + \lambda I)^{-1} S^* f_j^* + (C + \lambda I)^{-1} S^* f_j^* - (C + \lambda I)^{-1} S^* f^*\right]$$
$$=(C + \lambda I)^{-1/2} S^* (L - L_j)(L_j + \lambda I)^{-1} f_j^* + (C + \lambda I)^{-1/2} S^* (f_j^* - f^*)$$
$$=(C + \lambda I)^{-1/2} S^* (L - L_j)(L_j + \lambda I)^{-1} L^r L^{-r} f_j^* + (C + \lambda I)^{-1/2} S^* (f_j^* - f^*).$$

Since $\|(C + \lambda I)^{-1/2} S^*\| \leq 1$, $\|L\| \leq \kappa^2$, $\|C - C_j\| = \|L - L_j\|$ and $\Delta_{f_j} = \|f_j^* - f^*\|$, from Assumption 2, we have

$$\|(C + \lambda I)^{1/2}(\boldsymbol{w}_{j,\lambda} - \boldsymbol{w}_\lambda)\| \leq \frac{\kappa^{2r} R \Delta_{\mathfrak{D}_j}}{\lambda} + \Delta_{f_j}. \tag{28}$$

We then decompose the local variance

$$\boldsymbol{w}_{\mathfrak{D}_j,\lambda} - \boldsymbol{w}_{j,\lambda}$$
$$=(C_{\mathfrak{D}_j} + \lambda I)^{-1} S_{\mathfrak{D}_j}^* \boldsymbol{y}_{\mathfrak{D}_j} - (C_{\mathfrak{D}_j} + \lambda I)^{-1} S_j^* f_j^* + (C_{\mathfrak{D}_j} + \lambda I)^{-1} S_j^* f_j^* - (C_j + \lambda I)^{-1} S_j^* f_j^*$$
$$=(C_{\mathfrak{D}_j} + \lambda I)^{-1}(C_j + \lambda I)^{1/2}(C_j + \lambda I)^{-1/2}(S_{\mathfrak{D}_j}^* \boldsymbol{y}_{\mathfrak{D}_j} - S_j^* f_j^*) + [(C_{\mathfrak{D}_j} + \lambda I)^{-1} - (C_j + \lambda I)^{-1}] S_j^* f_j^*$$
$$=(C_{\mathfrak{D}_j} + \lambda I)^{-1}(C_j + \lambda I)^{1/2}(C_j + \lambda I)^{-1/2}(S_{\mathfrak{D}_j}^* \boldsymbol{y}_{\mathfrak{D}_j} - S_j^* f_j^*) + (C_{\mathfrak{D}_j} + \lambda I)^{-1}(C_j - C_{\mathfrak{D}_j})\boldsymbol{w}_{j,\lambda}$$
$$=(C_{\mathfrak{D}_j} + \lambda I)^{-1}(C_j + \lambda I)^{1/2}\left[(C_j + \lambda I)^{-1/2}(S_{\mathfrak{D}_j}^* \boldsymbol{y}_{\mathfrak{D}_j} - S_j^* f_j^*) + (C_j + \lambda I)^{-1/2}(C_j - C_{\mathfrak{D}_j})\boldsymbol{w}_{j,\lambda}\right].$$

and it holds

$$(C + \lambda I)^{1/2}(\boldsymbol{w}_{\mathfrak{D}_j,\lambda} - \boldsymbol{w}_{j,\lambda})$$
$$=(C + \lambda I)^{1/2}(C_{\mathfrak{D}_j} + \lambda I)^{-1}(C_j + \lambda I)^{1/2}\Big[(C_j + \lambda I)^{-1/2}(S_{\mathfrak{D}_j}^* \boldsymbol{y}_{\mathfrak{D}_j} - S_j^* f_j^*)$$
$$+ (C_j + \lambda I)^{-1/2}(C_j - C_{\mathfrak{D}_j})\boldsymbol{w}_{j,\lambda}\Big] \tag{29}$$
$$=(C + \lambda I)^{1/2}(C_j + \lambda I)^{-1/2}(C_j + \lambda I)^{1/2}(C_{\mathfrak{D}_j} + \lambda I)^{-1/2}(C_{\mathfrak{D}_j} + \lambda I)^{-1/2}(C_j + \lambda I)^{1/2}$$
$$\Big[(C_j + \lambda I)^{-1/2}(S_{\mathfrak{D}_j}^* \boldsymbol{y}_{\mathfrak{D}_j} - S_j^* f_j^*) + (C_j + \lambda I)^{-1/2}(C_j - C_{\mathfrak{D}_j})\boldsymbol{w}_{j,\lambda}\Big].$$

Due to Assumption 2 and $\|L_j\| \leq \kappa^2$, we obtain

$$\|\boldsymbol{w}_{j,\lambda}\|_K = \|(L_j + \lambda I)^{-1} L_j f_j^*\| = \|(L_j + \lambda I)^{-1} L_j L_j^r L_j^{-r} f_j^*\| \leq \kappa^{2r} \|L^{-r} f_j^*\| \leq \kappa^{2r} R. \tag{30}$$

Thus, substituting equation 30 to equation 29, using Lemma 3 and Lemma 4, for any $\delta \in (0, 1/2)$, we have with the probability $1 - 2\delta$

$$\|(C + \lambda I)^{1/2}(\boldsymbol{w}_{\mathfrak{D}_j,\lambda} - \boldsymbol{w}_{j,\lambda})\|$$
$$\leq \mathcal{P}_{\mathfrak{D}_j,\lambda}\sqrt{1 + \frac{\Delta_{\mathfrak{D}_j}}{\lambda}}\left(2B\kappa\mathcal{A}_{\mathfrak{D}_j,\lambda}\log\frac{2}{\delta} + 2\kappa(\kappa + 1)\mathcal{A}_{\mathfrak{D}_j,\lambda}\log\frac{2}{\delta}\kappa^{2r}R\right) \tag{31}$$
$$\leq 2\kappa(B + 2\kappa^3 R)\sqrt{1 + \frac{\Delta_{\mathfrak{D}_j}}{\lambda}}\mathcal{P}_{\mathfrak{D}_j,\lambda}\mathcal{A}_{\mathfrak{D}_j,\lambda}\log\frac{2}{\delta}.$$

Applying equation 28 and equation 31 to equation 27, we prove the result.

$\square$

**Theorem 5** (Detailed version of Theorem 2)**.** *For any* $\delta \in (0,1)$*, under Assumption 2, with the probability at least* $1 - \delta$*, the federated error holds*

$$\|\bar{f}^t_{\mathcal{D},\lambda} - f_{\mathcal{D},\lambda}\|_2 \le C_2 \Upsilon^t \sum_{j=1}^m p_j \sqrt{1 + \frac{\Delta_{\mathfrak{D}_j}}{\lambda}} \left( 2\mathcal{R}_{\mathfrak{D}_j,\lambda} + \frac{(1 + \mathcal{R}_{\mathfrak{D}_j,\lambda})\Delta_{\mathfrak{D}_j}}{\lambda} \right) \left( \mathcal{A}_{\mathfrak{D}_j,\lambda} \log \frac{2}{\delta} + \frac{\Delta_{\mathfrak{D}_j}}{\lambda} + \Delta_{f_j} \right).$$

$$(32)$$

*where* $C_2 = 2\kappa(B + 2\kappa^3 R)/(1 - \beta)$*,* $\beta = \lambda_{max}((C + \lambda)^{-1/2}(C - C_{\mathcal{D}})(C + \lambda)^{-1/2})$ *and* $\mathcal{A}_{\mathfrak{D}_j,\lambda} = \frac{1}{\sqrt{\lambda}|\mathfrak{D}_j|} + \sqrt{\frac{\mathcal{N}(\lambda)}{|\mathfrak{D}_j|}}$.

*Proof.* Substituting equation 26 and equation 17 to Theorem 1, with the probability $1 - 2\delta$, we obtain the federated error

$$\|\bar{f}^t_{\mathcal{D},\lambda} - f_{\mathcal{D},\lambda}\|_2$$

$$\le \Upsilon^t \left\| (C + \lambda I)^{1/2} (\bar{w}^0_{\mathcal{D},\lambda} - w_{\mathcal{D},\lambda}) \right\|_K$$

$$\le \Upsilon^t \sum_{j=1}^m p_j \mathcal{P}_{\mathcal{D},\lambda} \left( 2\mathcal{R}_{\mathfrak{D}_j,\lambda} + \frac{(1 + \mathcal{R}_{\mathfrak{D}_j,\lambda})\Delta_{\mathfrak{D}_j}}{\lambda} \right) \left\| (C + \lambda I)^{1/2} (w_{\mathfrak{D}_j,\lambda} - w_\lambda) \right\|_K$$

$$\le \Upsilon^t \sum_{j=1}^m p_j \mathcal{P}_{\mathcal{D},\lambda} \left( 2\mathcal{R}_{\mathfrak{D}_j,\lambda} + \frac{(1 + \mathcal{R}_{\mathfrak{D}_j,\lambda})\Delta_{\mathfrak{D}_j}}{\lambda} \right) \left( C_1 \sqrt{1 + \frac{\Delta_{\mathfrak{D}_j}}{\lambda}} \mathcal{P}_{\mathfrak{D}_j,\lambda} \mathcal{A}_{\mathfrak{D}_j,\lambda} \log \frac{2}{\delta} + \frac{\kappa^2 R \Delta_{\mathfrak{D}_j}}{\lambda} + \Delta_{f_j} \right)$$

$$\le \Upsilon^t \sum_{j=1}^m \frac{C_1 p_j}{1 - \beta} \sqrt{1 + \frac{\Delta_{\mathfrak{D}_j}}{\lambda}} \left( 2\mathcal{R}_{\mathfrak{D}_j,\lambda} + \frac{(1 + \mathcal{R}_{\mathfrak{D}_j,\lambda})\Delta_{\mathfrak{D}_j}}{\lambda} \right) \left( \mathcal{A}_{\mathfrak{D}_j,\lambda} \log \frac{2}{\delta} + \frac{\Delta_{\mathfrak{D}_j}}{\lambda} + \Delta_{f_j} \right).$$

$$(33)$$

The last step is due to Lemma 2. $\qquad \square$

### 7.5.2 ESTIMATING CENTRALIZED EXCESS RISK

The generalization analysis for the centralized model (the exact KRR) is standard Caponnetto & De Vito (2007); Smale & Zhou (2007), but the existing work imposed a strict assumption $r \in [1/2, 1]$ on the kernel space, which assumes the ideal estimator belongs to the kernel space $f^* \in \mathcal{H}_K$. Here, we relax this strict assumption to $r > 0$ but still obtain the identical optimal learning rates for the centralized excess risk bounds.

**Proposition 6.** *Under Assumption 2, for* $\delta \in (0, 1/2)$*, the following bounds hold with the probability at least* $1 - 2\delta$

$$\|f_{\mathcal{D},\lambda} - f^*\|_2 \le C_1 \mathcal{P}^{1/2}_{\mathcal{D},\lambda} \mathcal{A}_{\mathcal{D},\lambda} \log \frac{2}{\delta} + R\lambda^r, \qquad (34)$$

*where* $C_1 = 2\kappa \left( B + 2\kappa^3 R \right)$*.*

*Proof.* The excess risk term can be divided into two parts: variance and bias.

$$\|f_{\mathcal{D},\lambda} - f^*\| \le \|f_{\mathcal{D},\lambda} - f_\lambda\| + \|f_\lambda - f^*\|. \qquad (35)$$

Using Cauchy's inequality, Lemma 3 and Lemma 4, for $\delta \in (0, 1/2)$, with the probability at least $1 - 2\delta$ we have

$$\|f_{\mathcal{D},\lambda} - f_\lambda\|_2$$

$$=\|S(C_{\mathcal{D}} + \lambda I)^{-1} S_{\mathcal{D}}^* \boldsymbol{y}_{\mathcal{D}} - S(C_{\mathcal{D}} + \lambda I)^{-1} S^* f^* + S(C_{\mathcal{D}} + \lambda I)^{-1} S^* f^* - S(C + \lambda I)^{-1} S^* f^*\|_2$$

$$=\|S(C_{\mathcal{D}} + \lambda I)^{-1} (C + \lambda I)(C + \lambda I)^{-1/2} (C + \lambda I)^{-1/2} (S_{\mathcal{D}}^* \boldsymbol{y}_{\mathcal{D}} - S^* f^*)$$
$$+ S(C_{\mathcal{D}} + \lambda I)^{-1} (C + \lambda I)(C + \lambda I)^{-1/2} (C + \lambda I)^{-1/2} (C - C_{\mathcal{D}})(C + \lambda I)^{-1} S^* f^*\|_2$$

$$=\|S(C_{\mathcal{D}} + \lambda I)^{-1/2} (C_{\mathcal{D}} + \lambda I)^{-1/2} (C + \lambda I)^{1/2} (C + \lambda I)^{-1/2} (S_{\mathcal{D}}^* \boldsymbol{y}_{\mathcal{D}} - S^* f^*)$$
$$+ S(C_{\mathcal{D}} + \lambda I)^{-1/2} (C_{\mathcal{D}} + \lambda I)^{-1/2} (C + \lambda I)^{1/2} (C + \lambda I)^{-1/2} (C - C_{\mathcal{D}})(C + \lambda I)^{-1} S^* f^*\|_2$$

$$\leq 2B\kappa \log \frac{2}{\delta} \mathcal{P}_{\mathcal{D},\lambda}^{1/2} \mathcal{A}_{\mathcal{D},\lambda} + 2\kappa(\kappa + 1) \log \frac{2}{\delta} \mathcal{P}_{\mathcal{D},\lambda}^{1/2} \mathcal{A}_{\mathcal{D},\lambda} \|\boldsymbol{w}_\lambda\|_K$$

$$\leq 2\kappa \left(B + 2\kappa^3 R\right) \log \frac{2}{\delta} \mathcal{P}_{\mathcal{D},\lambda}^{1/2} \mathcal{A}_{\mathcal{D},\lambda}.$$

$$(36)$$

The last step is due $\|\boldsymbol{w}_\lambda\|_K = \|(L + \lambda I)^{-1} L f^*\| = \|(L + \lambda I)^{-1} L L^r L^{-r} f^*\| \leq \kappa^{2r} R$ due to Assumption 2.

The identity $A(A + \lambda I)^{-1} = I - \lambda(A + \lambda I)^{-1}$ holds for $\lambda > 0$ and $A$ the bounded self-adjoint positive operator. Then, under Assumption 2, it holds

$$\|f_\lambda - f^*\|_2$$
$$=\|(L + \lambda I)^{-1} L f^* - f^*\| = \|((L + \lambda I)^{-1} L - I) f^*\| = \|\lambda(L + \lambda I)^{-1} f^*\|$$
$$=\|\lambda^r \lambda^{1-r} (L + \lambda I)^{-(1-r)} (L + \lambda I)^{-r} L^r L^{-r} f^*\|$$
$$\leq \lambda^r \|\lambda^{1-r} (L + \lambda I)^{-(1-r)}\| \|(L + \lambda I)^{-r} L^r\| \|L^{-r} f^*\|$$
$$\leq R\lambda^r.$$

$$(37)$$

Substituting equation 36 and equation 37 to equation 35, we prove the result. $\qquad\square$

### 7.6 Excess Risk Bounds for FedNewton

*Proof of Theorem 3.* In the homogeneous setting, we have $\Delta_{\mathfrak{D}_j} = 0$ and $\Delta_{f_j} = 0$. Thus, under Assumption 2, from equation 32 and equation 34, it holds

$$\|\bar{f}_{\mathcal{D},\lambda}^t - f^*\|_2 \leq \|\bar{f}_{\mathcal{D},\lambda}^t - f_{\mathcal{D},\lambda}\|_2 + \|f_{\mathcal{D},\lambda} - f_{\mathcal{D},\lambda}\|_2$$

$$\leq \boldsymbol{O}\left(\Upsilon^t \sum_{j=1}^{m} p_j \mathcal{R}_{\mathfrak{D}_j,\lambda} \mathcal{A}_{\mathfrak{D}_j,\lambda} \log \frac{2}{\delta} + \mathcal{A}_{\mathcal{D},\lambda} \log \frac{2}{\delta} + R\lambda^r\right).$$

$$(38)$$

If $|\mathfrak{D}_j| > 29(\kappa^2 + 1) \log(1/\delta)/\lambda$, we have $\Upsilon < 1$. Otherwise, $\Upsilon \geq 1$.

From equation 10 and equation 22, under Assumption 1, with the probability at least $1 - 3\delta$, we have

$$\mathcal{R}_{\mathfrak{D}_j,\lambda} \mathcal{A}_{\mathfrak{D}_j,\lambda}$$

$$=\boldsymbol{O}\left(\left(\frac{1}{\lambda|\mathfrak{D}_j|} + \sqrt{\frac{1}{\lambda|\mathfrak{D}_j|}}\right) \log \frac{2}{\delta} \times \left(\frac{1}{|\mathfrak{D}_j|\sqrt{\lambda}} + \sqrt{\frac{\mathcal{N}(\lambda)}{|\mathfrak{D}_j|}}\right)\right)$$

$$=\boldsymbol{O}\left((|\mathfrak{D}_j|^{-2}\lambda^{-1.5} + |\mathfrak{D}_j|^{-1.5}\lambda^{-1-0.5\gamma} + |\mathfrak{D}_j|^{-1.5}\lambda^{-1} + |\mathfrak{D}_j|^{-1}\lambda^{-0.5-0.5\gamma}) \log \frac{2}{\delta}\right)$$

$$=\boldsymbol{O}\left((|\mathfrak{D}_j|^{-2}\lambda^{-1.5} + |\mathfrak{D}_j|^{-1.5}\lambda^{-1-0.5\gamma} + |\mathfrak{D}_j|^{-1}\lambda^{-0.5-0.5\gamma}) \log \frac{2}{\delta}\right).$$

The relationships between $\lambda$ and $|\mathfrak{D}_j|$ affects the value of $\mathcal{R}_{\mathfrak{D}_j,\lambda} \mathcal{A}_{\mathfrak{D}_j,\lambda}$.

$$\mathcal{R}_{\mathfrak{D}_j,\lambda} \mathcal{A}_{\mathfrak{D}_j,\lambda} = \log \frac{2}{\delta} \begin{cases} \boldsymbol{O}(|\mathfrak{D}_j|^{-2}\lambda^{-1.5}), & \text{if } \lambda < \boldsymbol{O}(|\mathfrak{D}_j|^{\frac{1}{\gamma-1}}). \\ \boldsymbol{O}(|\mathfrak{D}_j|^{-1.5}\lambda^{-1-0.5\gamma}), & \text{if } \Omega(|\mathfrak{D}_j|^{\frac{1}{\gamma-1}}) \leq \lambda < \boldsymbol{O}(|\mathfrak{D}_j|^{-1}). \\ \boldsymbol{O}(|\mathfrak{D}_j|^{-1}\lambda^{-0.5-0.5\gamma}), & \text{if } \lambda \geq \Omega(|\mathfrak{D}_j|^{-1}). \end{cases}$$

By setting $\lambda = |\mathcal{D}|^{\frac{-1}{2r+\gamma}}$ and $2r + \gamma \geq 1$, we have

$$
\mathcal{R}_{\mathfrak{D}_j,\lambda}\mathcal{A}_{\mathfrak{D}_j,\lambda} = \log\frac{2}{\delta}
\begin{cases}
\boldsymbol{O}\left(|\mathfrak{D}_j|^{-2}|\mathcal{D}|^{\frac{1.5}{2r+\gamma}}\right), & \text{if } |\mathfrak{D}_j| \lesssim |\mathcal{D}|^{\frac{1-\gamma}{2r+\gamma}}. \\
\boldsymbol{O}\left(|\mathfrak{D}_j|^{-1.5}|\mathcal{D}|^{\frac{1+0.5\gamma}{2r+\gamma}}\right), & \text{if } |\mathcal{D}|^{\frac{1-\gamma}{2r+\gamma}} \lesssim |\mathfrak{D}_j| \lesssim |\mathcal{D}|^{\frac{1}{2r+\gamma}}. \\
\boldsymbol{O}\left(|\mathfrak{D}_j|^{-1}|\mathcal{D}|^{\frac{1+\gamma}{4r+2\gamma}}\right), & \text{if } |\mathfrak{D}_j| \gtrsim |\mathcal{D}|^{\frac{1}{2r+\gamma}}.
\end{cases}
\tag{39}
$$

and

$$
\mathcal{A}_{\mathcal{D},\lambda} = |\mathcal{D}|^{\frac{1-4r-2\gamma}{4r+\gamma}} + |\mathcal{D}|^{\frac{-r}{2r+\gamma}} \leq 2|\mathcal{D}|^{\frac{-r}{2r+\gamma}}.
\tag{40}
$$

Substituting equation 39 and equation 40 to equation 38, we have

$$
\|\bar{f}_{\mathcal{D},\lambda}^t - f^*\|_2
$$

$$
\lesssim |\mathcal{D}|^{\frac{-r}{2r+\gamma}} \log\frac{2}{\delta} + \Upsilon^t \log^2\frac{2}{\delta} \sum_{j=1}^m p_j
\begin{cases}
|\mathfrak{D}_j|^{-2}|\mathcal{D}|^{\frac{1.5}{2r+\gamma}}, & \text{if } |\mathfrak{D}_j| \lesssim |\mathcal{D}|^{\frac{1-\gamma}{2r+\gamma}} \\
|\mathfrak{D}_j|^{-1.5}|\mathcal{D}|^{\frac{1+0.5\gamma}{2r+\gamma}}, & \text{if } |\mathcal{D}|^{\frac{1-\gamma}{2r+\gamma}} \lesssim |\mathfrak{D}_j| \lesssim |\mathcal{D}|^{\frac{1}{2r+\gamma}} \\
|\mathfrak{D}_j|^{-1}|\mathcal{D}|^{\frac{1+\gamma}{4r+2\gamma}}, & \text{if } |\mathcal{D}|^{\frac{1}{2r+\gamma}} \lesssim |\mathfrak{D}_j| \lesssim |\mathcal{D}|^{\frac{2r+\gamma+1}{4r+2\gamma}} \\
|\mathcal{D}|^{\frac{-r}{2r+\gamma}}, & \text{if } |\mathfrak{D}_j| \gtrsim |\mathcal{D}|^{\frac{2r+\gamma+1}{4r+2\gamma}}
\end{cases}
\tag{41}
$$

where

$$
\begin{cases}
t = 0, \Upsilon \geq 1, & \text{if } |\mathfrak{D}_j| \lesssim |\mathcal{D}|^{\frac{1}{2r+\gamma}} \\
t > 0, \Upsilon^t \lesssim \left(\frac{|\mathcal{D}|^{\frac{1}{2r+\gamma}}}{|\mathfrak{D}_j|}\right)^{0.5t}, & \text{otherwise.}
\end{cases}
\tag{42}
$$

Note that, $\Upsilon = 2\sum_{j=1}^m p_j \mathcal{P}_{\mathfrak{D}_j,\lambda}\mathcal{R}_{\mathfrak{D}_j,\lambda} \lesssim \sum_{j=1}^m p_j\mathcal{R}_{\mathfrak{D}_j,\lambda}$. When $|\mathfrak{D}_j| \gtrsim |\mathcal{D}|^{\frac{2r+\gamma+1}{4r+2\gamma}}$, we thus have $\mathcal{R}_{\mathfrak{D}_j,\lambda} \lesssim \sqrt{\frac{1}{\lambda|\mathfrak{D}_j|}} \lesssim |\mathcal{D}|^{\frac{1-2r-\gamma}{8r+4\gamma}}$. $\qquad\square$

*Proof of Theorem 4.* Under Assumption 2, from equation 32 and equation 34, it holds

$$
\|\bar{f}_{\mathcal{D},\lambda}^t - f^*\|_2 \leq \|\bar{f}_{\mathcal{D},\lambda}^t - f_{\mathcal{D},\lambda}\|_2 + \|f_{\mathcal{D},\lambda} - f_{\mathcal{D},\lambda}\|_2
$$

$$
\leq \boldsymbol{O}\left(\Upsilon^t \sum_{j=1}^m p_j \sqrt{1 + \frac{\Delta_{\mathfrak{D}_j}}{\lambda}}\left(2\mathcal{R}_{\mathfrak{D}_j,\lambda} + \frac{(1+\mathcal{R}_{\mathfrak{D}_j,\lambda})\Delta_{\mathfrak{D}_j}}{\lambda}\right)\left(\mathcal{A}_{\mathfrak{D}_j,\lambda}\log\frac{2}{\delta} + \frac{\Delta_{\mathfrak{D}_j}}{\lambda} + \Delta_{f_j}\right) + \mathcal{A}_{\mathcal{D},\lambda}\log\frac{2}{\delta} + R\lambda^r\right).
$$

Let $\lambda = |\mathcal{D}|^{\frac{-1}{2r+\gamma}}$ and $2r + \gamma \geq 1$. When $|\mathfrak{D}_j| \leq \boldsymbol{O}(|\mathcal{D}|^{\frac{1}{2r+\gamma}})$, we have $\frac{1}{\lambda|\mathfrak{D}_j|} \geq \sqrt{\frac{1}{\lambda|\mathfrak{D}_j|}} \geq 1$ and $\mathcal{R}_{\mathfrak{D}_j,\lambda} \lesssim \frac{1}{\lambda|\mathfrak{D}_j|} + \sqrt{\frac{1}{\lambda|\mathfrak{D}_j|}} \lesssim \frac{1}{\lambda|\mathfrak{D}_j|}$ from equation 10. Thus,

$$
\|\bar{f}_{\mathcal{D},\lambda}^t - f^*\|_2
$$

$$
\leq \boldsymbol{O}\left(\Upsilon^t \sum_{j=1}^m p_j\left(1 + \frac{\Delta_{\mathfrak{D}_j}}{\lambda}\right)^{1.5}\left(\mathcal{R}_{\mathfrak{D}_j,\lambda}\mathcal{A}_{\mathfrak{D}_j,\lambda}\log\frac{2}{\delta} + \frac{\mathcal{R}_{\mathfrak{D}_j,\lambda}\Delta_{\mathfrak{D}_j}}{\lambda} + \mathcal{R}_{\mathfrak{D}_j,\lambda}\Delta_{f_j}\right) + \mathcal{A}_{\mathcal{D},\lambda}\log\frac{2}{\delta} + R\lambda^r\right)
$$

$$
\leq \boldsymbol{O}\left(\Upsilon^t \sum_{j=1}^m p_j\left(1 + \frac{\Delta_{\mathfrak{D}_j}}{\lambda}\right)^{1.5}\left(\mathcal{R}_{\mathfrak{D}_j,\lambda}\mathcal{A}_{\mathfrak{D}_j,\lambda}\log\frac{2}{\delta} + \frac{\Delta_{\mathfrak{D}_j}}{\lambda^2|\mathfrak{D}_j|} + \frac{\Delta_{f_j}}{\lambda|\mathfrak{D}_j|}\right) + \mathcal{A}_{\mathcal{D},\lambda}\log\frac{2}{\delta} + R\lambda^r\right)
$$

$$
\leq \boldsymbol{O}\left(\Upsilon^t \sum_{j=1}^m p_j\left(1 + \frac{\Delta_{\mathfrak{D}_j}}{\lambda}\right)^{1.5}\left(\mathcal{R}_{\mathfrak{D}_j,\lambda}\mathcal{A}_{\mathfrak{D}_j,\lambda} + \frac{|\mathcal{D}|^{\frac{2}{2r+\gamma}}}{|\mathfrak{D}_j|}\Delta_{\mathfrak{D}_j} + \frac{|\mathcal{D}|^{\frac{1}{2r+\gamma}}}{|\mathfrak{D}_j|}\Delta_{f_j}\right)\log\frac{2}{\delta} + |\mathcal{D}|^{\frac{-r}{2r+\gamma}}\right).
$$

When $|\mathfrak{D}_j| \geq \Omega(|\mathcal{D}|^{\frac{1}{2r+\gamma}})$, we have $\mathcal{R}_{\mathfrak{D}_j,\lambda} \lesssim \sqrt{\frac{1}{\lambda|\mathfrak{D}_j|}} \leq 1$, $\mathcal{A}_{\mathfrak{D}_j,\lambda} \lesssim |\mathfrak{D}_j|^{-1/2}|\mathcal{D}|^{\frac{\gamma/2}{2r+\gamma}}$ and

$$\|\bar{f}_{\mathcal{D},\lambda}^t - f^*\|_2$$

$$\leq \boldsymbol{O}\left( \Upsilon^t \sum_{j=1}^m p_j \sqrt{1 + \frac{\Delta_{\mathfrak{D}_j}}{\lambda}} \left( \mathcal{R}_{\mathfrak{D}_j,\lambda} + \frac{\Delta_{\mathfrak{D}_j}}{\lambda} \right) \left( \mathcal{A}_{\mathfrak{D}_j,\lambda} \log \frac{2}{\delta} + \frac{\Delta_{\mathfrak{D}_j}}{\lambda} + \Delta_{f_j} \right) + |\mathcal{D}|^{\frac{-r}{2r+\gamma}} \right)$$

$$\leq \boldsymbol{O}\left( \Upsilon^t \sum_{j=1}^m p_j \sqrt{1 + \frac{\Delta_{\mathfrak{D}_j}}{\lambda}} \left( \mathcal{R}_{\mathfrak{D}_j,\lambda}\mathcal{A}_{\mathfrak{D}_j,\lambda} + |\mathcal{D}|^{\frac{1}{2r+\gamma}}\Delta_{\mathfrak{D}_j} + \Delta_{f_j} + \frac{|\mathcal{D}|^{\frac{\gamma+2}{4r+2\gamma}}}{\sqrt{|\mathfrak{D}_j|}}\Delta_{\mathfrak{D}_j} + |\mathcal{D}|^{\frac{2}{2r+\gamma}}\Delta_{\mathfrak{D}_j}^2 \right. \right.$$

$$\left. \left. + |\mathcal{D}|^{\frac{1}{2r+\gamma}}\Delta_{\mathfrak{D}_j}\Delta_{f_j} \right) \log \frac{2}{\delta} + |\mathcal{D}|^{\frac{-r}{2r+\gamma}} \right)$$

$$\leq \boldsymbol{O}\left( \Upsilon^t \sum_{j=1}^m p_j \sqrt{1 + \frac{\Delta_{\mathfrak{D}_j}}{\lambda}} \left( \mathcal{R}_{\mathfrak{D}_j,\lambda}\mathcal{A}_{\mathfrak{D}_j,\lambda} + |\mathcal{D}|^{\frac{1}{2r+\gamma}}\Delta_{\mathfrak{D}_j} + \Delta_{f_j} + |\mathcal{D}|^{\frac{2}{2r+\gamma}}\Delta_{\mathfrak{D}_j}^2 + |\mathcal{D}|^{\frac{1}{2r+\gamma}}\Delta_{\mathfrak{D}_j}\Delta_{f_j} \right) \log \frac{2}{\delta} \right.$$

$$\left. + |\mathcal{D}|^{\frac{-r}{2r+\gamma}} \right)$$

$$\leq \boldsymbol{O}\left( \Upsilon^t \sum_{j=1}^m p_j \sqrt{1 + \frac{\Delta_{\mathfrak{D}_j}}{\lambda}} \left( \mathcal{R}_{\mathfrak{D}_j,\lambda}\mathcal{A}_{\mathfrak{D}_j,\lambda} + (1 + |\mathcal{D}|^{\frac{1}{2r+\gamma}}\Delta_{\mathfrak{D}_j})(|\mathcal{D}|^{\frac{1}{2r+\gamma}}\Delta_{\mathfrak{D}_j} + \Delta_{f_j}) \right) \log \frac{2}{\delta} + |\mathcal{D}|^{\frac{-r}{2r+\gamma}} \right).$$

Combing with equation 39, we complete the proof

$$\|\bar{f}_{\mathcal{D},\lambda}^t - f^*\|_2 \lesssim \Upsilon^t \sum_{j=1}^m p_j \sqrt{1 + \frac{\Delta_{\mathfrak{D}_j}}{\lambda}}(\aleph_j + \Pi_j) \log^2 \frac{2}{\delta} + |\mathcal{D}|^{\frac{-r}{2r+\gamma}} \log \frac{2}{\delta}.$$

Here, $\aleph_j$ and $\Pi_j$ have different values w.r.t local sample size

$$\aleph_j = \begin{cases} |\mathfrak{D}_j|^{-2}|\mathcal{D}|^{\frac{1.5}{2r+\gamma}}, & \text{if } |\mathfrak{D}_j| \lesssim |\mathcal{D}|^{\frac{1-\gamma}{2r+\gamma}} \\ |\mathfrak{D}_j|^{-1.5}|\mathcal{D}|^{\frac{1+0.5\gamma}{2r+\gamma}}, & \text{if } |\mathcal{D}|^{\frac{1-\gamma}{2r+\gamma}} \lesssim |\mathfrak{D}_j| \lesssim |\mathcal{D}|^{\frac{1}{2r+\gamma}} \\ |\mathfrak{D}_j|^{-1}|\mathcal{D}|^{\frac{1+\gamma}{4r+2\gamma}}, & \text{if } |\mathcal{D}|^{\frac{1}{2r+\gamma}} \lesssim |\mathfrak{D}_j| \lesssim |\mathcal{D}|^{\frac{2r+\gamma+1}{4r+2\gamma}} \\ |\mathcal{D}|^{\frac{-r}{2r+\gamma}}, & \text{if } |\mathfrak{D}_j| \gtrsim |\mathcal{D}|^{\frac{2r+\gamma+1}{4r+2\gamma}}, \end{cases}$$

and

$$\Pi_j = \begin{cases} \frac{|\mathcal{D}|^{\frac{2}{2r+\gamma}}}{|\mathfrak{D}_j|}\Delta_{\mathfrak{D}_j} + \frac{|\mathcal{D}|^{\frac{1}{2r+\gamma}}}{|\mathfrak{D}_j|}\Delta_{f_j}, & \text{if } |\mathfrak{D}_j| \lesssim |\mathcal{D}|^{\frac{1}{2r+\gamma}} \\ (1 + |\mathcal{D}|^{\frac{1}{2r+\gamma}}\Delta_{\mathfrak{D}_j})(\Delta_{f_j} + |\mathcal{D}|^{\frac{1}{2r+\gamma}}\Delta_{\mathfrak{D}_j}), & \text{if } |\mathfrak{D}_j| \gtrsim |\mathcal{D}|^{\frac{1}{2r+\gamma}}. \end{cases}$$

$\square$