# OpenReview forum: "Efficient Newton-type Federated Learning with Non-IID Data"
_ICLR.cc/2025/Conference — ICLR 2025 Conference Withdrawn Submission_

### Official Review · Reviewer_Syn3 · 2024-11-01

**Soundness:** 4
**Presentation:** 3
**Contribution:** 4
**Rating:** 6
**Confidence:** 3

**Summary:**

The paper proposes FedNewton that uses Newton-type optimization to achieve faster convergence and better accuracy, when data across clients is heterogeneous. In typical federated learning methods like FedAvg and FedProx, only first-order gradient information is used, which often fails to perform well when data distributions differ significantly among clients. FedNewton, on the other hand, leverages second-order information by approximating Hessians, allowing it to handle these data heterogeneity better. The paper includes strong theoretical guarantees on generalization and convergence rates, and it breaks down the federated error into components that account for covariate shift and concept shift—where differences in data distributions and target relationships across clients make global model training challenging. With the help of second-order updates, FedNewton improves communication efficiency while achieving better accuracy, given in experiments on synthetic and real-world datasets.

**Strengths:**

This paper is very well written, and I enjoyed reading it. The strength of this paper is

(1) Well-motivated problem.
(2) Strong theoretical analysis.

**Weaknesses:**

I have a few queries about it.

(1) I think I did not understand the formulation properly, which is why I have doubts about it.
Lines 091-093 state, “By averaging the local models, the simplest federated method only communicates once, known as Distributed Kernel Ridge Regression (DKRR) with the closed-form solution.” If it has closed-form solutions, what is needed for Newton’s method?

(2) What are the limitations of the proposed methods? In particular, in which cases would the First-order methods work better than FedNewton?

(3) This paper has solid theoretical foundations, but I observed that the experiments are very limited. The authors should also focus on the experiments. Authors should consider other state-of-the-art Newton-type methods (FedDANE, FedNEW, FedNL, etc.). If the author could compare the proposed model with some of the state-of-the-art empirically, then the paper would be stronger.

(4) Is it possible to train a Neural network-based model? Or is it limited to Kernel ridge regression only?

(5) Is there any empirical evidence of  FedNewton in the presence of model heterogeneity? It would be nice to see how robust this model is in the presence of model heterogeneity.

**Questions:**

Asked in weakness

---

### Official Review · Reviewer_nr7W · 2024-11-03

**Soundness:** 3
**Presentation:** 4
**Contribution:** 2
**Rating:** 3
**Confidence:** 3

**Summary:**

This paper focuses on second-order horizontal FL in the Kernal Ridge Regression setting. The paper proposed the FedNewton algorithm, which first aggregates the local gradient and then aggregates the local Hessian inverse multiples aggregated gradient for the update. The paper provides rigorous proof for the proposed algorithm on its convergence rate. Additional numerical results on synthetic and MNIST datasets demonstrate that the proposed algorithm outperforms existing algorithms.

**Strengths:**

Solid theoretical analysis:
1. The paper provides a solid theoretical analysis of the convergence of the KRR problem with the proposed algorithm.
2. The results on both homogeneous and heterogeneous distributions demonstrate how the algorithm's convergence rate is affected by the data distribution.

Clarity: The notations and assumptions are clearly defined. The theoretical results are also well-stated.

**Weaknesses:**

1. Limited use case:
    1. The analysis only works for KRR problem, and cannot be generalized to other problems/models.
    2. The algorithm requires computing the full local gradient and local Hessian of the problem, which is computationally expensive. This limits the usage of the algorithm to large-scale optimization.
    3. The algorithm requires full client-participation and full-batch computation, which might be impractical in the FL setting.

2. Insufficient numerical result:
    1. The numerical experiments are only on synthetic and LIBSVM datasets, while other datasets should be considered (e.g., LEAF benchmark)
    2. The comparison to SOTA results is missing. The SOTA accuracy for FL algorithms on the MNIST dataset is much higher (e.g., 98% in FedDyn Acar et al. 2021.)

3. Poor organization: The reviewer understands the theoretical contribution is the major focus of the paper. However, the authors should provide a reference to the specific appendix and a summary of the experimental results in the main contribution.

**Questions:**

Please address the weaknesses above, especially 1.1, 1.2, and 1.3.

---

### Official Review · Reviewer_BBJc · 2024-11-03

**Soundness:** 2
**Presentation:** 3
**Contribution:** 2
**Rating:** 5
**Confidence:** 3

**Summary:**

This paper proposes a second-order method for Federated Learning (FL), specifically tailored for simple models. The authors provide a thoughtful analysis of the convergence and generalization bounds of the proposed method. Experimental results indicate faster convergence and a slight increase in test accuracy.

**Strengths:**

1. Originality: This paper introduces a Newton-type Federated Learning method that shares both first-order and second-order knowledge across heterogeneous data.

2. Quality: The analysis of the convergence and generalization bounds of FedNewton, particularly in the context of the Kernel Ridge Regression (KRR) model, is well thought out.

**Weaknesses:**

1. Lack of Comprehensive Baselines: The comparison is limited to traditional methods such as FedAvg, FedProx, and FeDKRR, which restricts the ability to verify the novelty and superiority of the proposed method. The authors should compare their work with established Newton-type Federated Learning methods in experiments,  e.g., [1], [2], [3].

2. Limited Applicability: The proposed method appears to be applicable only to very simple machine learning models like KRR. The algorithm and analysis are rooted in this model, which may not generalize to more complex tasks. Moreover, the practical implementation of Newton-type methods in deep learning models is challenging due to the high computational cost of Hessian calculations. The authors emphasize their contribution by comparing their method with traditional KRR methods and older first-order FL methods (e.g., FedAvg and FedProx), which may not adequately showcase the advantages of their approach.

3. Marginal Performance Improvement: The experimental results suggest that the proposed method offers only a slight performance enhancement. Could you please provide additional analysis or ablation studies to demonstrate that the performance improvements are due to their methodological contribution rather than hyperparameter tuning.

Ref:

[1] Elgabli A, Issaid C B, Bedi A S, et al. FedNew: A communication-efficient and privacy-preserving Newton-type method for federated learning[C]//International conference on machine learning. PMLR, 2022: 5861-5877.

[2] Dal Fabbro N, Dey S, Rossi M, et al. SHED: A Newton-type algorithm for federated learning based on incremental Hessian eigenvector sharing[J]. Automatica, 2024, 160: 111460.

[3] Li J, Liu Y, Wang W. FedNS: A Fast Sketching Newton-Type Algorithm for Federated Learning[C]//Proceedings of the AAAI Conference on Artificial Intelligence. 2024, 38(12): 13509-13517.

**Questions:**

1. Please clarify the differences and contributions of FedNewton compared to recent state-of-the-art Newton-type FL methods. A thorough literature review would help identify relevant baseline studies.

2. Is it possible to provide experimental results involving deep learning models, like the convolutional neural network (CNN), and take into account SOTA Newton-type FL methods as baselines? Could you please discuss the challenges when applying the proposed method to deep learning models and how you might address these challenges.

---

### Note · Authors · 2024-11-20

I have read and agree with the venue's withdrawal policy on behalf of myself and my co-authors.